# Super-resolution microscopy reveals majorly mono- and dimeric presenilin1/γ-secretase at the cell surface

Abril Angélica Escamilla-Ayala[1,2], Ragna Sannerud[1,2], Magali Mondin[3], Karin Poersch[4], Wendy Vermeire[1,2], Laura Paparelli[1,2,5], Caroline Berlage[6], Marcelle Koenig[7], Lucia Chavez-Gutierrez[2,8], Maximilian H Ulbrich[9,10], Sebastian Munck[2,5], Hideaki Mizuno[11], Wim Annaert[1,2]*

[1]Laboratory for Membrane Trafficking, VIB-KU Leuven Center for Brain and Disease Research, Leuven, Belgium; [2]Department of Neurosciences, KU Leuven, Leuven, Belgium; [3]Bordeaux Imaging Center, UMS 3420, CNRS-University of Bordeaux, US4 INSERM, Bordeaux, France; [4]Faculty of Biology, University of Freiburg, Freiburg, Germany; [5]VIB Bio Imaging Core, Leuven, Belgium; [6]Einstein Center for Neurosciences, NeuroCure Cluster of Excellence, Charité-Universitätsmedizin Berlin, Berlin, Germany; [7]PicoQuant GmbH, Berlin, Germany; [8]Laboratory of Proteolytic Mechanisms in Neurodegeneration, VIB-KU Leuven Center for Brain and Disease Research, Leuven, Belgium; [9]Institute of Internal Medicine IV, Medical Center of the University of Freiburg, Freiburg, Germany; [10]BIOSS Centre for Biological Signaling Studies, University of Freiburg, Freiburg, Germany; [11]Laboratory of Biomolecular Network Dynamics, Biochemistry, Molecular and Structural Biology Section, KU Leuven, Heverlee, Belgium

*For correspondence:
wim.annaert@kuleuven.vib.be

**Abstract** γ-Secretase is a multi-subunit enzyme whose aberrant activity is associated with Alzheimer's disease and cancer. While its structure is atomically resolved, γ-secretase localization in the membrane in situ relies mostly on biochemical data. Here, we combined fluorescent tagging of γ-secretase subunits with super-resolution microscopy in fibroblasts. Structured illumination microscopy revealed single γ-secretase complexes with a monodisperse distribution and in a 1:1 stoichiometry of PSEN1 and nicastrin subunits. In living cells, sptPALM revealed PSEN1/γ-secretase mainly with directed motility and frequenting 'hotspots' or high track-density areas that are sensitive to γ-secretase inhibitors. We visualized γ-secretase association with substrates like amyloid precursor protein and N-cadherin, but not with its sheddases ADAM10 or BACE1 at the cell surface, arguing against pre-formed megadalton complexes. Nonetheless, in living cells PSEN1/γ-secretase transiently visits ADAM10 hotspots. Our results highlight the power of super-resolution microscopy for the study of γ-secretase distribution and dynamics in the membrane.

## Introduction

The γ-secretase complex is an intramembrane cleaving protease (i-CLiP) formed by the catalytic presenilin (PSEN), nicastrin (NCT), anterior-pharynx defective 1 (APH1) and presenilin enhancer 2 (PEN2) (*De Strooper, 2003*; *Kimberly et al., 2003*). It cleaves more than a hundred type-I transmembrane substrates (*Jurisch-Yaksi et al., 2013*) including Notch (important for development and cancer), ErbB4 (involved in breast cancer), and the Amyloid Precursor Protein (APP), whose cleavage product is the toxic amyloid-β, a major hallmark in Alzheimer's disease (*Annaert and De Strooper, 2002*;

*Selkoe and Hardy, 2016*). To fully explore the potential of γ-secretase as a drug target in the fields of neurodegeneration and cancer, a detailed understanding of its structure and biology is required.

The complex is assembled in a 1:1:1:1 ratio (*Sato et al., 2007*) in early biosynthetic compartments, a process thought to be mediated through recycling from the Golgi to the endoplasmic reticulum (ER) (*Spasic et al., 2007*). Whereas overexpression of any subunit results in its ER jamming, the knock-out (KO) of a single subunit prevents complex assembly and destabilizes remaining subunits, all together underscoring that all are required for complex assembly (*Steiner et al., 2002*; *Luo et al., 2003*). A mature functional γ-secretase complex is characterized by glycosylated NCT (*Yu et al., 2000*) and the endoproteolysis and heterodimerization of PSEN N- and C-termini (*Luo et al., 2003*). This form is long-lived as opposed to unassembled subunits that are rapidly degraded (*De Strooper and Annaert, 2010*).

While enormous progress has been made in the past years to unveil γ-secretase protein structure at atomic resolution (*Bai et al., 2015b*), we are still lacking direct evidence to reveal γ-secretase distribution and dynamics in membranes of living cells. Only microscopy techniques like FLIM-FRET and molecular dynamics simulations based on the γ-secretase 3D-cryo structure have helped to better understand conformational changes γ-secretase undergoes for substrate recognition and cleaving mechanisms (*Wolfe, 2019*). Less is known about the lateral nanoscale organization of γ-secretase at the plasma membrane (PM), and these insights are still based on biochemical and fractionation studies. Lipidome analysis of immunoprecipitated γ-secretase identified cholesterol and phosphatidylcholine molecules, which were also identified in γ-secretase 3D-cryo structures (*Yang et al., 2019*). These findings could support the association of γ-secretase with cholesterol- and sphingomyelin-enriched domains called lipid rafts. However, to date, most of these data rely on the biochemical characterization of detergent-resistant membranes and flotation gradient centrifugation (*Vetrivel et al., 2004*; *Hur et al., 2008*). Likewise, co-immunoprecipitation and gel filtration approaches found γ-secretase associated with high molecular weight, up to megadalton, complexes that include the cognate sheddases, ADAM10 and BACE1 (*Chen et al., 2015*; *Liu et al., 2019*). However, this lateral nanoscale organization of γ-secretase has thus far not been corroborated with imaging approaches for long because of the lack of resolution.

Given the diffraction limit of light, conventional widefield and confocal microscopy are limited to ~200 nm lateral resolution. Thus, proteins distributed at a distance below this limit cannot be distinguished from one another. The limited resolution significantly jeopardizes co-localization analysis commonly used to assess protein-protein associations. This has dramatically changed in the past decade with the introduction of super-resolution and quantitative microscopy (*Lee et al., 2017*; *Schermelleh et al., 2019*). It delivered not only unique close-ups of subcellular structures but also powerful tools to study biological processes, including protein nanodistribution at the single-molecule level and in their native context (*Zuidscherwoude et al., 2015*; *Nicovich et al., 2017*; *Lagache et al., 2018*). Thanks to Structured Illumination Microscopy (SIM), Stimulated Emission Depletion (STED) microscopy and Single-Molecule Localization Microscopy (SMLM), the lateral resolution has been increased up to ~100 nm for SIM and ~20 nm for STED and SMLM. These and other advanced microscopy techniques have revealed the importance of nanodomain organization of proteins and lipids, allowing now to study their impact on downstream signaling (*Nakajo et al., 2010*; *Hastie et al., 2013*; *Tobin et al., 2018*). Under living conditions, these nanodomains can be seen as dynamic hotspots for protein recruitment or confinement, as revealed by single-particle tracking coupled to photo-activated localization microscopy (sptPALM) (*Manley et al., 2008*) in molecular dynamics maps (*Rossier et al., 2012*; *Nair et al., 2013*; *Constals et al., 2015*; *Bademosi et al., 2016*; *Padmanabhan et al., 2019*).

In this study, we combined SIM, molecular counting by photobleaching (*Ulbrich and Isacoff, 2007*; *Arant and Ulbrich, 2014*) and sptPALM to advance the current biochemical knowledge on γ-secretase. We focus on PSEN1/γ-secretase complexes not only for their relevance for the Alzheimer's disease and cancer (*Haapasalo and Kovacs, 2011*; *Jurisch-Yaksi et al., 2013*; *Voytyuk et al., 2018*), but also because it is particularly abundant at the PM making it more accessible to different microscopy techniques (*Sannerud et al., 2016*). We show for the first time PSEN1/γ-secretase stoichiometry at the PM at single-molecule resolution and provide evidence of its majorly mono- and dimeric distribution in living cells. sptPALM identified dynamic hotspots where PSEN1/γ-secretase is recruited, which are influenced by γ-secretase inhibitor (GSI) treatments, denoting a yet unknown dynamic nanodomain organization. Importantly, and in contrast to earlier studies

(*Chen et al., 2015*; *Liu et al., 2019*), our analysis does not provide support for preformed megadalton-sized associations of PSEN1/γ-secretase with sheddases. Rather, SMLM suggests that hotspots are frequented by the different secretases in a transient or temporally regulated manner. Therefore, our study sets out new criteria for γ-secretase research at nanoscale resolution.

## Results

### γ-Secretase PSEN1 and NCT subunits are in a 1:1 stoichiometry at the cell surface

Studies related to the complex formation and stoichiometry of γ-secretase have been essentially biochemical in nature. Here, we set out to validate the biochemical model of γ-secretase subunit stoichiometry by direct visualization using super-resolution imaging approaches. We selected PSEN1 for being the most studied PSEN and for being more abundantly expressed at the cell surface (*Sannerud et al., 2016*). We introduced GFP or mEOS3.2 on PSEN1 and SNAP-tag on NCT for dual-color fluorescence microscopy. To ensure that the introduced tags did not interfere with complex assembly and maturation, we first tested each of them in single KO backgrounds. For both, PSEN1 endoproteolysis and mature glycosylation of NCT was observed (*Figure 1—figure supplement 1A, B*), as well as broad localization in different subcellular compartments (*Figure 1—figure supplement 1C*). This underscores that the introduced tags do not interfere with normal maturation of subunits and complex formation, which is in agreement with previous reports on PSEN and NCT tagging (*Kaether et al., 2006*; *Cui et al., 2015*; *Sannerud et al., 2016*). For the single-molecule visualization of both tagged subunits in one γ-secretase complex in situ, we next engineered a triple KO (tKO; NCT$^{-/-}$, PSEN1$^{-/-}$, PSEN2$^{-/-}$) mouse embryonic fibroblast (MEF) cell line using CRISPR/Cas9 gene editing. These tKO MEFs were subsequently used to stably re-introduce NCT-SNAP and GFP-PSEN1 to generate rescued cell lines with two tagged γ-secretase subunits (*Figure 1A*). To obtain balanced levels of both subunits, we labeled NCT-SNAP with the far-red fluorescent cell-permeable SNAP-SiR ligand and used Fluorescence-Activated Cell Sorting (FACS) to select four populations depending on the relative levels of either PSEN1 or NCT (*Figure 1—figure supplement 1D*). We selected P8 for further experiments, which (i) had the highest levels of mature glycosylated NCT-SNAP and PSEN1 heterodimers, but (ii) exhibited no apparent accumulations of full-length PSEN1 (indicating non-incorporation into complexes), (iii) showed PEN2 stabilization, and (iv) normalized processing of APP-CTF as a direct substrate of γ-secretase (*Figure 1—figure supplement 1E*). Blue native PAGE of a 0.5% n-Dodecyl β-D-maltoside (DDM) cell extract displayed the expected ~440 kDa high molecular weight complex (HMWC), as previously reported (*Kimberly et al., 2003*; *Fraering et al., 2004*) and indicating proper assembly of the tagged subunits into a mature complex (*Figure 1—figure supplement 1F*). To confirm the presence of mature tagged complexes at the cell surface, we used aminolipid-coated superparamagnetic iron oxide nanoparticles (SPIONs) to isolate highly purified PMs (*Figure 1—figure supplemental 1G*; *Tharkeshwar et al., 2017*). Subsequent pull-down with GFP antibodies from CHAPSO extracted PMs (which preserves γ-secretase complex integrity), followed by SDS-PAGE and quantitative western blotting showed that GFP-PSEN1-NTF co-immunoprecipitated with NCT-SNAP and PSEN1-CTF in ratios that are similar to those from the input (*Figure 1B*). Blue native PAGE of DDM-extracted PMs further confirmed the integrity of the tagged complex (*Figure 1C*). Lastly, quantitative western blotting, using recombinant γ-secretase as a standard, gave a GFP-PSEN1/NCT-SNAP ratio in tKO samples comparable to the one observed in WT and single KO (sKO) GFP-PSEN1 (*Figure 1D*). Confocal imaging revealed co-localized subunits at the cell surface, including membrane ruffles (*Figure 1E*), and in LAMP1-positive organelles (*Figure 1—figure supplement 1H*). Thus, the tKO NCT-SNAP/GFP-PSEN1 rescued MEFs demonstrated a normal assembly, maturation and localization of the tagged subunits into γ-secretase complexes, indistinguishable from WT cells. Moreover, the proper trafficking of the complex to the cell surface, a major resident compartment for PSEN1/γ-secretase, allowed us now to study its cell surface distribution in unprecedented detail using super-resolution microscopy.

Given the broad subcellular localization of PSEN1 not only as a subunit of γ-secretase in endosomes and at the cell surface, but also in an unassembled state in the ER (*Annaert et al., 1999*; *Réchards et al., 2003*), we turned to supported PM sheets to clearly visualize only the fraction of PSEN1/γ-secretase at the cell surface (*Chaney and Jacobson, 1983*; *Paparelli et al., 2016*). By

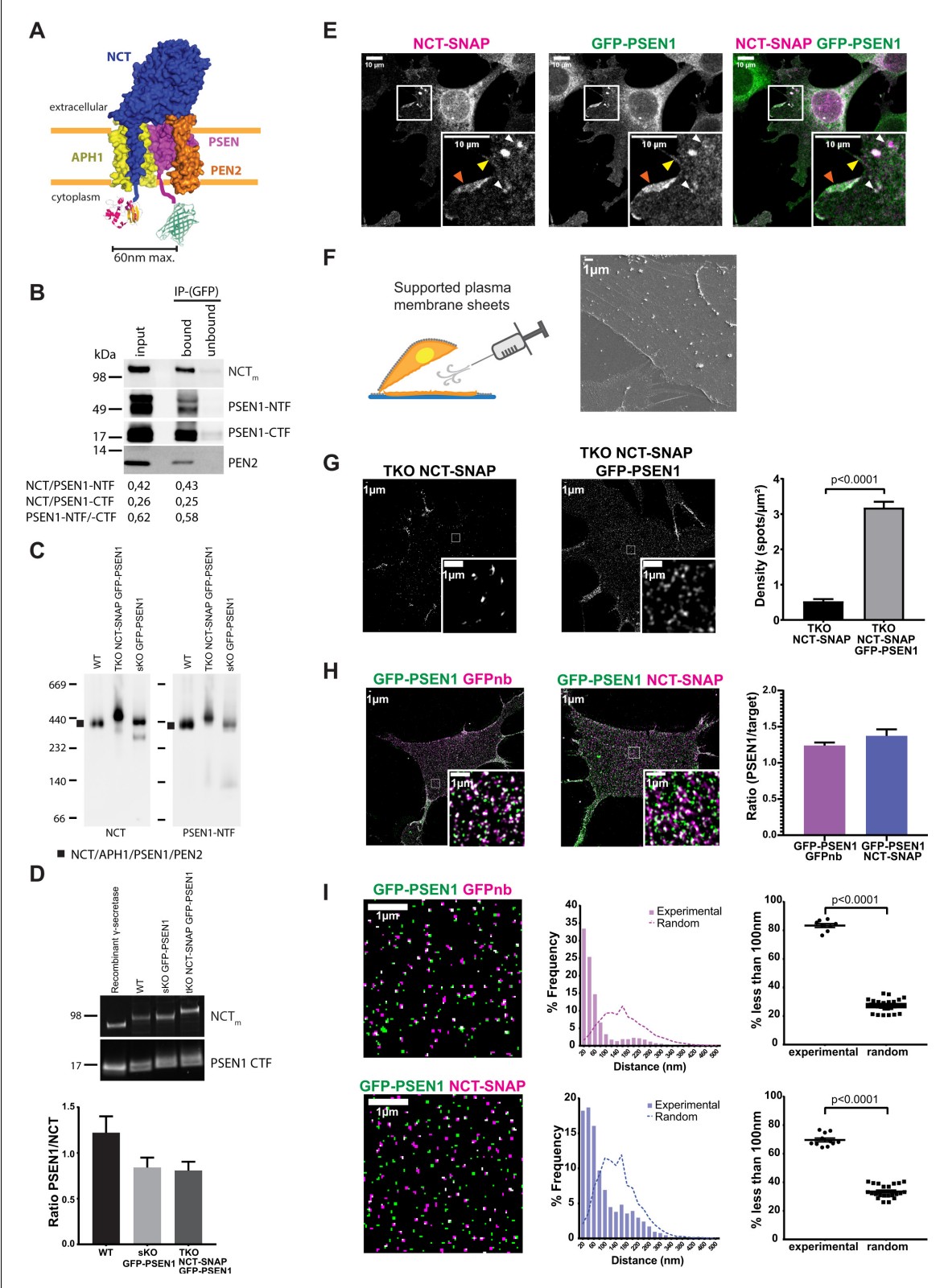

**Figure 1.** Visualizing the stoichiometry of γ-secretase subunits. (**A**) Scheme of double-tagged γ-secretase. Both tags face the cytoplasm with the SNAP-tag and a linker fused to the C-terminus of NCT, and GFP with a linker to the N-terminus of PSEN1. Maximum distance between both tags is theoretically ~60 nm. (**B**) Co-immunoprecipitation of CHAPSO extracts of SPION isolated PM fractions using anti-GFP antibodies. Western blot shows an almost complete depletion of the extract for γ-secretase components in a ratio (NCT/PSEN1-NTF (0.43), NCT/PSEN1-CTF (0.25) and PSEN1-NTF /-

*Figure 1 continued on next page*

Figure 1 continued

CTF (0.58)) similar to those from input (0.42, 0.26, 0.62, respectively). (C) Blue native PAGE and western blot analysis of PM fractions of WT, NCT-SNAP/GFP-PSEN1 rescued tKO and GFP-PSEN1 rescued sKO MEFs. All samples show a band of around 440 kDa corresponding to the full γ-secretase complex (*Fraering et al., 2004*). The faint lower band corresponds to NCT/PSEN1-NTF/APH1A as DDM extraction slightly affects the stability of the complex as previously reported (*Spasic et al., 2007*; *Fraering et al., 2004*). (D) Quantitative western blot of SPION isolated PM fractions normalized to known amounts of purified recombinant γ-secretase. The PSEN1/NCT ratio is close to one for all samples. n = 4 replicates (E) Confocal microscopy of SiR-*labelled* NCT-SNAP/GFP-PSEN1 rescued tKO MEFs showing co-localization at the cell surface (yellow arrowhead), including membrane ruffles (orange arrowhead), and LAMPI-positive vesicles (white arrowheads). Scale bar = 10 μm (F) (Left panel) Schematic representation of supported PM sheet preparation. (Right panel) Scanning Electron Microscopy (SEM) of PM sheets of GFP-PSEN1 rescued sKO MEFs showing the basal PM attached to the coverslip. Some cytoskeleton can be seen still attached. Scale bar = 1 μm (G) SIM on PM sheets of NCT-SNAP and NCT-SNAP/GFP-PSEN1 rescued tKO MEFs showing a dramatically reduced NCT-SNAP spot density when tKO MEFs are only rescued with NCT-SNAP (mean ± SEM, NCT-SNAP n = 8 cells; NCT-SNAP/GFP-PSEN1 n = 14 cells). Scale bar = 1 μm (H) SIM image of PM sheets of NCT-SNAP/GFP-PSEN1 rescued tKO MEFs labeled with GFPnb-Atto647n (left panel) or SNAP-SiR (middle panel) showing similar spot densities in both channels (right panel) (mean ± SEM, GFPnb n = 8 cells; NCT n = 11 cells). (I) Masks of SIM PM sheets of NCT-SNAP/GFP-PSEN1 rescued tKO MEFs with either GFPnb-Atto647n (upper panels) or SNAP-SiR (lower panels). Scale bar = 1 μm. Histograms show the distribution of nearest-neighbor distances of either GFPnb to PSEN1 or NCT to PSEN1 spot centroids. (Left panels) Dot plot summarizing nearest-neighbor distances below 100 nm. Random distances were calculated from unpaired experimental data. Each dot represents one cell. Comparison analysis by two-tail Mann-Whitney test (mean ± SEM, GFPnb n = 8 cells; NCT n = 11 cells). See *Figure 1—source data 1*.

The online version of this article includes the following source data and figure supplement(s) for figure 1:

**Source data 1.** Source Data for Nearest Neighbor Analysis.
**Figure supplement 1.** Biochemical characterization of MEF rescued cell lines.
**Figure supplement 2.** PM sheets controls.

coating cell monolayers with silica beads, the upper PM is rigidified and can be removed by shear force, leaving only the basal PM attached to the coverslip as shown by scanning electron microscopy (*Figure 1F*). Actin staining with Phalloidin-Alexa647 revealed that some cytoskeletal elements remained attached to the PM, but the PM is otherwise essentially devoid of other adhering organelles (*Figure 1—figure supplement 2A*). Although this method has been previously reported to not disturb native organization of molecules (*Paparelli et al., 2016*), we compared as a control PSEN1 organization at the basal PM of entire cells versus supported PM sheets in PSEN1 sKO MEFs stably rescued with mEOS3.2-PSEN1. Analysis of reconstructed PALM images confirmed that PM sheets do not majorly disturb the native organization of molecules as shown by the cluster radius and overall distribution of points (*Figure 1—figure supplement 2B*). Of note, these clusters shown by PALM do not represent an equal number of mEOS3.2-PSEN1 molecules, but result from the blinking property of mEOS3.2 giving rise to multiple localizations per molecule.

To further assure that different fluorescent tags do not have an impact on PSEN1/γ-secretase distribution at the PM, we analyzed PM sheets of MEF cell lines with different genetic backgrounds stably expressing fluorescently tagged-PSEN1. Widefield images were analyzed by the tessellation-based QuASIMoDOH algorithm, which measures slight changes in lateral density (*Paparelli et al., 2016*). This analysis demonstrated that tagged PSEN1/γ-secretase spots are randomly distributed, regardless of the genetic background and fluorescent tag (*Figure 1—figure supplement 2C*). Moreover, density analysis of SIM resolved spots showed a similar pattern in all cell lines, giving an average of 10 spots/μm$^2$ (*Figure 1—figure supplement 2D*). Importantly, SIM analysis demonstrated as well that supported PM sheets give a superior signal-to-noise ratio making this approach highly recommended for quantitative fluorescence microscopy of PM proteins and lipids.

We next analyzed PM sheets from tKO MEFs rescued with only NCT-SNAP or together with GFP-PSEN1 imaged by SIM. In the absence of PSEN expression, tKO NCT-SNAP MEFs had a close to 8-fold decreased NCT-SNAP density at the cell surface compared to tKO NCT-SNAP/GFP-PSEN1 MEFs (*Figure 1G*). These results demonstrate that the trafficking of unassembled subunits is strongly impaired when another subunit, in this case, PSEN1, is missing further consolidating the normal complex assembly in tKO NCT-SNAP/GFP-PSEN1 MEFs.

We then visualized NCT and PSEN1 subunits in the same γ-secretase complex at the PM. PM sheets from tKO NCT-SNAP-SiR/GFP-PSEN1 MEFs displayed most GFP-PSEN1 spots overlapping with or adjoining NCT-SNAP-SiR spots (*Figure 1H*, middle panel). The density ratio of NCT-SNAP:GFP-PSEN1 was indistinguishable from immunolabeling GFP-PSEN1 with GFPbooster-Atto647N (GFPnb) indicating a one to one stoichiometry (*Figure 1H*). This was confirmed by nearest-neighbor

analysis, wherein the distance to the nearest-neighbor is measured starting from the centroid coordinates of the spot. Herein, NCT-SNAP-SiR over GFP-PSEN1 revealed a peak at ~50 nm, that is slightly larger compared to the ~20 nm for GFP-PSEN1 over GFPbooster-Atto647N, but clearly distinct from a theoretical random distribution (*Figure 1I*). We noticed a fraction of the experimental population that follows the random distribution in both cases, representing about 16% for GFPnb and 30% for NCT of the total population. These likely correspond to GFPnb or NCT-SNAP that pair with GFP-PSEN1 of another complex that is not visible, either due to GFP photobleaching or immature chromophores. Indeed, the expected fraction of fluorescent GFP at the PM for a given GFP construct is ~80%, which is in line with our observed association to GFPnb or NCT-SNAP by GFP-PSEN1 (*Ulbrich and Isacoff, 2007*). As the resolution of SIM is limited to 100 nm, distances up to 100 nm are considered associations. The slightly higher paired distance for PSEN1-NCT compared to the dual-labeled GFP-PSEN1 may originate from the linkers used to fuse the SNAP-tag and GFP to NCT and PSEN1, respectively. Theoretically, they separate the tags to a maximum of ~60 nm (*Figure 1A*). Thus, the close apposition/co-localization of NCT and PSEN1 at the PM of fully rescued tKO MEFs confirms the biochemical predicted 1:1 ratio in γ-secretase complexes.

## PSEN1/γ-secretase is majorly mono and dimeric at the cell surface

As NCT-PSEN1 spots had almost a 1:1 ratio in a very short distance range (<50 nm), these spots likely correspond to one fluorophore each, thus representing one γ-secretase complex per spot. To determine whether this is indeed the case, we generated a NCT KO MEF line using CRISPR/Cas9. This line was subsequently stably rescued with both NCT-SNAP and NCT-GFP lentiviral vectors and after selection and quality control, PM sheets were imaged by SIM. If γ-secretase is monomeric, distances to the nearest-neighbor for differently labeled NCT molecules will be similar to a random distribution. On the other hand, if γ-secretase exists in higher-order multimers or clusters, we expect strong associations below 100 nm. Nearest-neighbor analysis demonstrated a close to random distribution, underscoring an essentially monomeric distribution of NCT molecules at the cell surface (*Figure 2A*). To further support this, we applied two independent approaches, first by counting the photobleaching steps of GFP-PSEN1 (*Figure 2B*), and second, by photon counting the number of fluorescent emitters of NCT-YFP at the cell surface (*Figure 2—figure supplement 1*; *Ulbrich and Isacoff, 2007*; *Ta et al., 2015*). Both approaches showed that the monomeric spots are the most abundant (30% to 35%), followed by dimeric units (22% to 30%), whereas higher complexes (trimers and above) are more rarely observed.

As fixatives may alter the normal distribution of complexes resulting in the co-incidence of multiple spots below the diffraction limit, we set out to explore this in living cells. Since γ-secretase complexes in living cells were very mobile and their paths cross each other, counting the photobleaching steps was impossible. Instead, we used the intensity of the complexes to estimate the number of fluorescently labeled subunits they contain. In tKO MEFs rescued with NCT-GFP and GFP-PSEN1, we could expect two GFP tags per γ-secretase complex, and therefore, the double intensity as for a single GFP tag, as in the case of a sKO GFP-PSEN1 cell line. Because the density of PSEN1/γ-secretase was too high to spatially separate individual molecules, we used a technique to reduce the molecule density while conserving the labeling stoichiometry, called PhotoGate (*Madl et al., 2010*; *Belyy et al., 2017*). This approach is based on fast photobleaching of a region of interest and partial re-population of the bleached area by fluorescent complexes diffusing from outside this region. Hence, this method only allows us to study mobile complexes. Since these mobile complexes were not exposed to the pre-bleaching pulse, their fluorescent labels are still intact allowing their intensity analysis at single-molecule resolution. After the initial photobleaching step, we observed diffusing fluorescent spots (*Figure 2C*; see *Video 1*). Some of the spots started with a high intensity and proceeded through an intermediate intensity level to complete bleaching, while others did not show the intermediate level (*Figure 2D*).

We compared intensity histograms of the beginning and the end of the movie, after which most GFPs are bleached and therefore, only complexes with a single functional GFP should be left (*Figure 2E*). The histogram of tKO NCT-GFP/GFP-PSEN1 MEFs showed a pronounced shoulder at twice the intensity of a single GFP that disappeared at the end of movie acquisition, meaning that the distribution shifted from complexes with one or two intact GFP tags to only one remaining GFP tag. In contrast, the distribution of spot intensities in sKO GFP-PSEN1 MEFs was similar at the beginning and the end of the movie. The histogram of spot intensities from NCT-GFP/GFP-PSEN1 MEFs

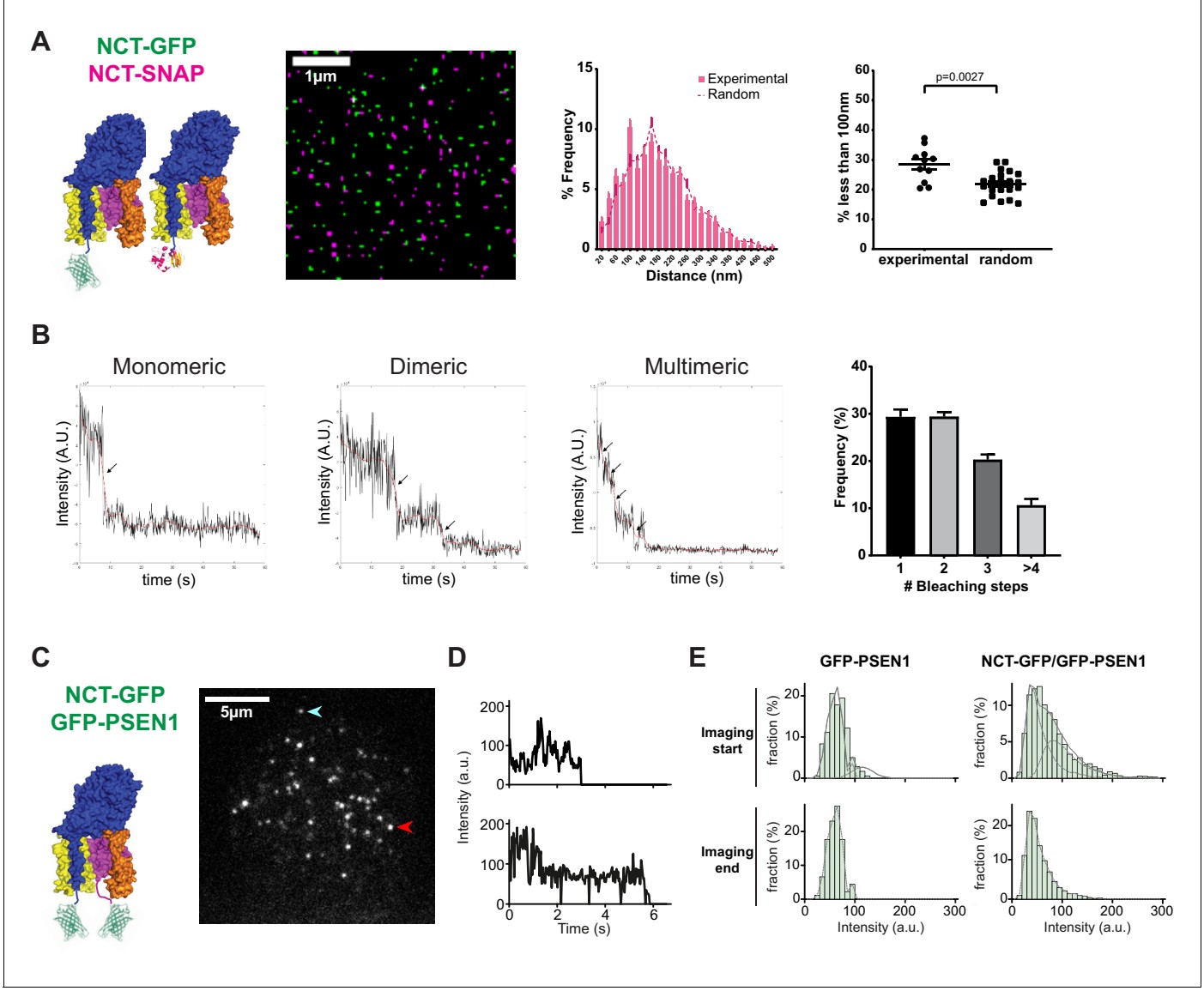

**Figure 2.** PSEN1/γ-secretase is monomeric and dimeric at the cell surface. (A) SIM analysis of PM sheets derived from NCT KO MEFs rescued with both NCT-GFP and NCT-SNAP. Mask of SIM images shows no to little color overlap. Nearest-neighbor distances distribution and respective dot plot of distances (<100 nm) show little differences from a random distribution. Comparison analysis by two-tail Mann-Whitney test (mean ± SEM, n = 11 cells). (B) Photobleaching step examples of monomeric, dimeric and multimeric GFP-PSEN1 spots in PM sheets. Histogram shows the distribution of photobleaching steps per cell (mean ± SEM, n = 4 cells). (C) Live-cell single-molecule intensity measurements. NCT-GFP/GFP-PSEN1 rescued tKO or GFP-PSEN1 rescued sKO MEFs imaged in TIRF post-photogate, to diminish the molecular density of the ROI. After bleaching, unbleached molecules diffuse to the bleached area. Scale bar = 5 µm (D) Intensity measurement of individual spots marked by an arrow in (C). (E) Comparison of GFP-PSEN1 with NCT-GFP/GFP-PSEN1 intensity histograms. (Upper panels) Histograms of the first 100 frames of the movies show two populations, one with the unitary intensity, and one with twice the unitary intensity (dashed curves) for NCT-GFP/GFP-PSEN1; the second population is absent in GFP-PSEN1. (Lower panels) Histograms of the last 100 frames of the movies showed depletion of double intensity population in NCT-GFP/GFP-PSEN1 after photobleaching during the recording, leaving only single intensity molecules in both cases. See *Figure 2—source data 1* and *Figure 2—source data 2*.

The online version of this article includes the following source data and figure supplement(s) for figure 2:

**Source data 1.** Source Data for Nearest Neighbor Analysis.
**Source data 2.** Source Data for Photobleaching analysis.
**Figure supplement 1.** Molecular counting by photon statistics (CoPS) on PM sheets derived from NCT-YFP rescued NCT KO MEFs.

**Video 1.** Molecular counting of GFP-tagged γ-secretase complexes. An ROI was prebleached by PhotoGate and molecules were allowed to diffuse into the ROI prior to imaging. (left panel-control) GFP-TRCP4a transfected cell shows diffraction–limited spots containing 4x GFP intensities as these molecules are organized as tetramers. (middle panel) A tKO NCT-GFP GFP-PSEN1 cell shows diffraction-limited spots of 2x GFP intensity (one each on NCT and PSEN1) reflecting the 1:1 NCT-PSEN1 subunit stoichiometry. (right panel) A sKO GFP-PSEN1 cell shows diffraction-limited spots of 1x GFP intensity as γ-secretase consists of only one PSEN1 molecule per complex. Color table shows fluorescence intensities (a.u). Time stamp shows seconds of recording.

https://elifesciences.org/articles/56679#video1

yields 42% of spots with two functional GFP and 58% with one functional GFP. Based on 20% of GFP that remain dark (likely due to misfolding) and that has been observed previously (*Ulbrich and Isacoff, 2007*; *Madl et al., 2010*), we would expect a slightly higher fraction of 67% of spots with two functional GFP. The lower fraction in our experiment can be explained by additional pre-bleaching of complexes during cell search and focusing, by a minor fraction of complexes that contain only one GFP, or a combination of both. Since in post-Golgi compartments, γ-secretase subunits exist essentially in a complex, we cannot exclude that unassembled, immature subunits from the PM-juxtaposed cortical ER diffuse into the focal plane, further explaining this pool of single GFP molecules. The ER harbors relatively most of the unassembled and immature subunits, further supporting this. Taking the different independent approaches together, our data indicate that under both fixed and live conditions, the majority of PSEN1/γ-secretase complexes are mono- or dimeric complexes, but not higher-order clusters.

## PSEN1/γ-secretase displays fast diffusion at the plasma membrane

Thus far, our data demonstrate a random distribution of single or dimeric PSEN1/γ-secretase complexes, but lacks kinetic insight in how these tetrameric complexes behave laterally in the PM of living cells. To address this, we stably rescued PSEN1 sKO MEFs with PSEN1 tagged with the photoswitchable fluorescent protein, mEOS3.2. This allows to perform sptPALM, with the restriction that only short tracks of few seconds can be captured. However, short tracks are compensated by an increase in the number of molecules sampled at a given time. We noticed that even in these short tracks, we could observe changes in PSEN1 lateral behavior. To appreciate which behavior is predominant in the population, we analyzed the different motilities depending on the distance travelled during a given time period such as immobile, confined, anomalous, Brownian or directed. This is represented by the mean squared displacement (MSD), whose fit $r^2 = t^\alpha + k$ gives us information on the type of motility by evaluating α (*Sibarita, 2014*). On average, PSEN1/γ-secretase showed anomalous motility (mean α = 0.844) (*Figure 3B*). However, when classifying individual tracks, most of them have a predominant directed motility mixed with other types of motility (*Figure 3C*). To further understand the dynamics of γ-secretase, we calculated the average diffusion coefficient by fitting the first 4 points of the MSD curve. This revealed an average diffusion coefficient of 0.15 μm/s$^2$ for PSEN1/γ-secretase (*Figure 3D*). Further analysis showed that most of the tracks are mobile (87%) with a small fraction being immobile (*Figure 3E*). The latter is defined by tracks that stay within an area smaller than the spatial resolution of the imaging system (~50 nm, D ~ 0.01 μm$^2$/s, $Log_{10}D = -2$) extracted from the value of time zero in the MSD.

The wide variety of diffusion coefficients in tracks could reflect different conformational states of γ-secretase, as shown previously for other proteins (*Suzuki et al., 2012*; *Constals et al., 2015*). Recent progress in understanding the structural biology of PSEN1/γ-secretase showed the existence of compact/semi-open conformations likely reflecting the active state of the enzyme versus the rigidifying effects of GSI (*Bai et al., 2015a*; *Bai et al., 2015b*; *Aguayo-Ortiz and Dominguez, 2018*; *Hitzenberger and Zacharias, 2019*; *Zhou et al., 2019*). To test if conformational changes affected PSEN1/γ-secretase diffusion at the PM, we treated mEOS3.2-PSEN1 MEFs with the GSIs DAPT and L-685,458 (InhX). DAPT 'locks' the γ-secretase structure in an intermediate state between the active and inactive conformation, while InhX is a transition-state analog that binds to the active site in a similar manner as the substrate APP (*Hitzenberger and Zacharias, 2019*), inducing a 'compact/semi-open' conformation (*Elad et al., 2015*). Exposure to GSI did not affect the MSD, diffusion

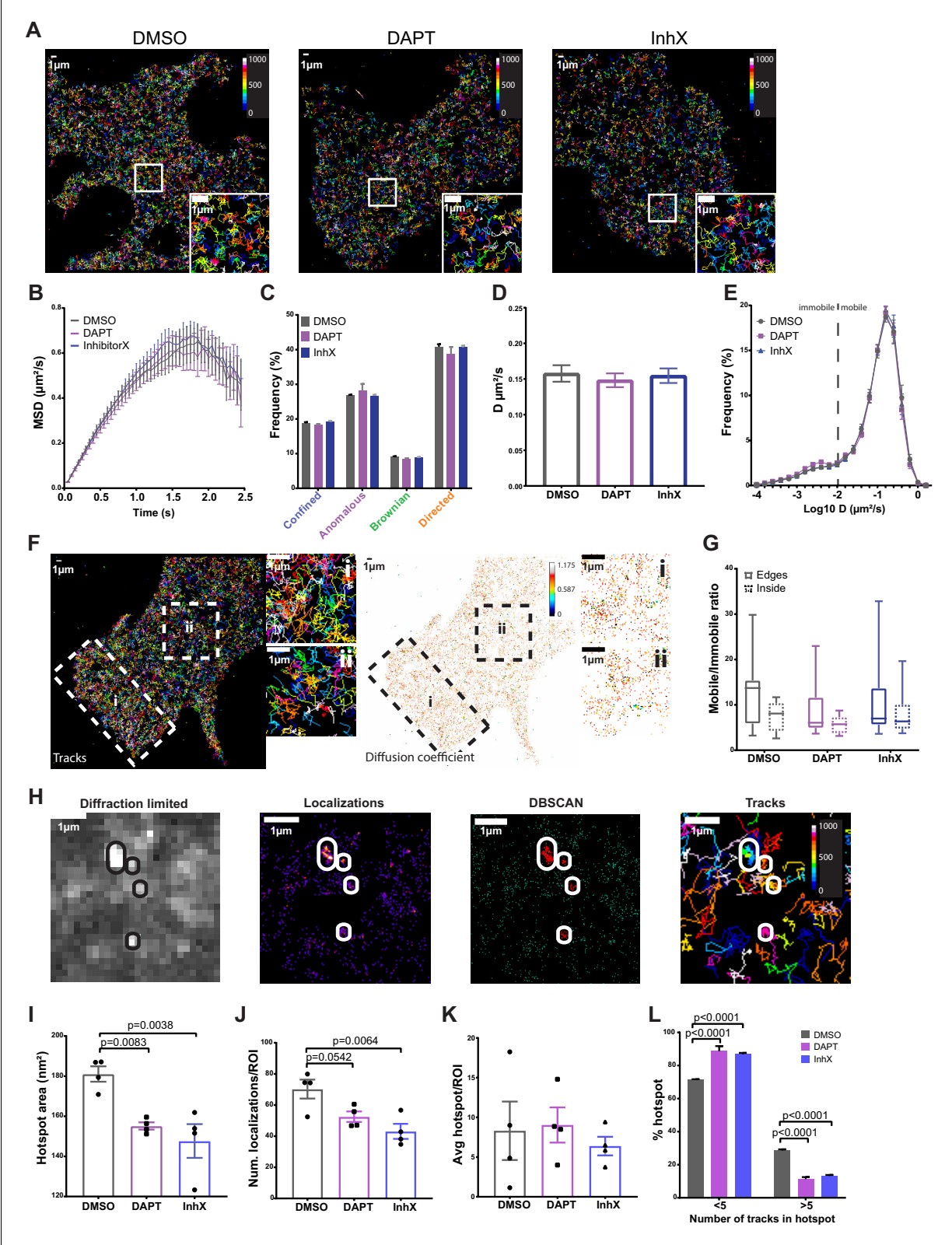

**Figure 3.** PSEN1/γ-secretase diffusion is fast at the cell surface and unaffected by GSI treatments, whereas hotspot areas display sensitivity to treatments. Rescued MEFs were treated for 1 hr at 37°C with 1 μm of either DAPT or InhX (L-685,458) prior to imaging. DMSO (1:1000) was used as a control. (**A**) Representative cells showing similar distribution of mEOS3.2-PSEN1 tracks of cells treated with GSI or control (Bar code = frame of track initiation). Scale bar = 1 μm (**B**) Plot with the MSD over time shows an average anomalous movement (α = 0.844), which is unchanged upon GSI

*Figure 3 continued on next page*

*Figure 3 continued*

treatment (mean ± SEM, n = 14 cells). (**C**) Comparison of each motility frequency in the absence (control) or presence of GSIs (mean ± SEM, n = 14). (**D**) Average diffusion coefficients are not different between control and GSI treated cells (mean ± SEM, n = 14 cells). (**E**) Frequency distributions of diffusion coefficients (as $Log_{10}$) identifies a major mobile and minor immobile pool. The dotted line marks the limit resolution of the microscope, rendering everything below it as immobile fraction. Only in the case of DAPT, the immobile pool is slightly increased (mean ± SEM, n = 14 cells). (**F**) Representative sptPALM image of mEOS3.2-PSEN1 tracks (left panel) and individual diffusion coefficient distribution (right panel, bar code = individual D) during ~1.5 min recording. Edge region marked by 'i' and boxed inside region by 'ii'. Scale bar = 1 μm (**G**) Mobile/immobile ratio from tracks either in cell edges or in the inner region. There is a non-significant trend towards more mobile tracks in the cell periphery that is abolished after GSI treatments (mean, min-max, DMSO n = 12 cells, DAPT n = 6 cells, InhX n = 11 cells). (**H**) Diffraction-limited projection (left panel) of mEOS-PSEN1 hotspots (encircled). sptPALM localizations (left-middle panel) analyzed by DBSCAN revealed hotspots (right-middle panel) consisting of overlapping tracks (right panel). For cluster analysis, four to eight ROIs of the same size per cell were defined and wherein cell edges were avoided. Scale bar = 1 μm (**I**) Hotspot area is significantly reduced in GSI treated cells. Comparison analysis by one-way ANOVA (mean ± SEM, n = 4 cells, 30 hotspots/cell). (**J**) Number of localizations per hotspot shows a trend towards less localizations per hotspot upon treatment with GSI. Comparison analysis by one-way ANOVA (mean ± SEM, n = 4 cells, 30 hotspots/cell). (**K**) Mean number of hotspots per ROI is variable but not different in control and GSI treated cells (mean ± SEM, n = 4 cells, 30 hotspots/cell). (**L**) Number of tracks per hotspot is reduced upon GSI treatment. Multiple comparisons by two-way ANOVA (mean ± SEM, n = 4 cells, 30 hotspots/cell). See *Figure 3—source data 1* and *Figure 3—source data 2*.

The online version of this article includes the following source data and figure supplement(s) for figure 3:

**Source data 1.** Source Data for Hotspot analysis.
**Source data 2.** Source Data for SPT analysis.
**Figure supplement 1.** Track length information.

coefficient or the overall distribution of $Log_{10}$ D (*Figure 3A–E*). We only found a slight increase of the immobile fraction upon DAPT treatment visible in the $Log_{10}$ histogram showing the diffusion coefficient of individual tracks (*Figure 3E*). Upon closer inspection of PSEN1/γ-secretase tracks, we observed a slight enrichment of tracks at the cell edge (*Figure 3F*). To analyze this further, we selected regions from the inner part of the cell (*Figure 3F,ii*) and the periphery of the cell (*Figure 3F,i*) and analyzed, according to its diffusion coefficient, the ratio of mobile to immobile tracks. The apparent 'enrichment' of tracks at the cell edge tended to correlate with an increased number of mobile tracks (*Figure 3G*). As we imaged in TIRF, the possibility arises that we include some molecules from the upper membrane near the edges, but this was maximally avoided by restricting the localization precision (16 nm ±3.7) during the tracking analysis. Nevertheless, because we are measuring the ratio of mobile/immobile tracks, the density of tracks is normalized. Our analysis thus suggests a higher presence of PSEN1/γ-secretase in the protruding or migrating edges of the cell and GSI treatment mildly reduced the distribution of mobile tracks at these edges.

Thus far, the spatial super-resolution analysis of PSEN1/γ-secretase does not support a clustered organization, reflecting its presence in high molecular weight complexes. Alternatively, these protease complexes might only associate transiently in a dynamic subdomain that cannot be revealed by spatial analysis in fixed cells. To identify such 'hot-spots', defined as areas with a higher frequency in encountering a PSEN1/γ-secretase complex passing through or halting, we analyzed the single-molecule localization data of the sptPALM using Density-Based Spatial Clustering Applications with Noise (DBSCAN) (*Ester, 1996*; *Figure 3H*). To avoid cell edges, four to eight squared ROIs of the same size per cell were analyzed for hotspots. We defined a hotspot in DBSCAN as an area with more than 18 localizations to avoid having only one immobile track (most tracks have only ten localizations (*Figure 3—figure supplement 1*)). Of note, we found that DBSCAN–defined hotspots correspond to high-intensity regions in a diffraction-limited projection of the sptPALM stack (*Figure 3H*), which allows also the analysis through masks used later in the study. Interestingly, the size (*Figure 3I*), but not the average number of hotspots (*Figure 3K*), became significantly reduced following treatment with GSIs. In addition, the localizations per hotspot showed a diminishing trend (*Figure 3J*). When examining the number of tracks per hotspot, fewer tracks were found, explaining the smaller hotspot area in GSI-treated cells (*Figure 3L*). These data suggest that locking PSEN1/γ-secretase conformation in an active-mimicking conformation restricts visiting these hotspots. One could hypothesize that a conformational change induced by GSIs affects the interaction with proteins and/or lipids thereby affecting the normal mobility and localization of PSEN1/γ-secretase to hotspot areas at the PM.

In summary, although GSI induce conformational compaction of PSEN1/γ-secretase, this does not strikingly affect the kinetics nor density of PSEN1/γ-secretase complexes at the cell surface. Furthermore, our sptPALM data reveal a highly dynamic PSEN1/γ-secretase complex across the PM with different types of motilities not associated with a conformational change induced by GSI. Alternatively, this diversity in motilities could reflect transient associations of the complex with lipids or proteins or distinct nanodomain organizations (*Sibarita, 2014*; *Bianchi et al., 2018*; *Martínez-Muñoz et al., 2018*). Interestingly, GSI did affect the visit of hotspot areas by PSEN1/γ-secretase, showing some relation with γ-secretase activity.

## PSEN1/γ-secretase associates with substrates but only transiently to sheddases

Given that our nearest-neighbor analysis on SIM spots enables us to visualize single γ-secretase complex associations at the PM, we decided next to investigate biochemically proven PSEN1/γ-secretase associations, such as with APP and N-cadherin, two surface localized substrates. Despite the lower density of endogenous APP at the cell surface, nearest-neighbor analysis demonstrated a significant association of PSEN1/γ-secretase with both substrates below a 100 nm distance, supporting in situ association with substrates (*Figure 4A,B*). We then extended this to ADAM10 and BACE1, two sheddases preceding γ-secretase in the dual processing of substrates. Of note, because of the lack of proper antibodies, we expressed SNAP-tagged ADAM10 or BACE1 in GFP-PSEN1 expressing sKO MEFs and limited our analysis only to those cells with the lowest expression to avoid overexpression artifacts. In contrast to substrates, nearest-neighbor analysis showed almost no difference compared to a random distribution analysis of the same spots (*Figure 4C,D*). Only a small fraction of ADAM10 (6% ± 0.4) showed association with PSEN1/γ-secretase. This is surprising as recent biochemical evidence pointed to the association of these sheddases with PSEN1/γ-secretase in long-lived megadalton (>5 MDa) complexes (*Chen et al., 2015*; *Liu et al., 2019*). Quantitative super-resolution microscopy, however, does not overall support this.

Nevertheless, given our previous finding in living conditions that PSEN1/γ-secretase hotspots could be linked to γ-secretase activity, we decided to compare diffraction-limited hotspots of mEOS3.2-PSEN1 with hotspots of ADAM10-SNAP and of BACE1-SNAP in living cells. However, hotspots of sheddases did not overlap with those of PSEN1/γ-secretase as shown by the very low Pearson's and Mander's coefficient (*Figure 5A–C*). As PSEN1 hotspots correspond to a high density of tracks, we next analyzed instead the association of individual PSEN1/γ-secretase tracks with ADAM10- or BACE1-SNAP hotspots (*Figure 5D*). Although there is a tendency of more PSEN1/γ-secretase tracks being associated with BACE1 hotspots (*Figure 5E*), they had high diffusion coefficients (D = 0.08 ± 0.01 μm²/s), indicating that these tracks represent PSEN1/γ-secretase merely passing by BACE1 hotspots without actually being associated to it (*Figure 5F*). On the contrary, PSEN1/γ-secretase tracks that frequent ADAM10 hotspots have decreased diffusion coefficients (D = 0.04 ± 0.02 μm²/s, *Figure 5F*). These tracks may, therefore, represent complexes that are slowing down to associate with ADAM10-frequented regions. As our nearest-neighbor analysis could not support significant close or direct associations, an alternative explanation might be that PSEN1/γ-secretase and ADAM10 frequent common nanodomains at the PM, the nature of which remains to be investigated.

## Discussion

Almost twenty years after the discovery of γ-secretase (*De Strooper, 2003*; *Kimberly et al., 2003*), our knowledge of the biology of this fascinating tetrameric complex remains scarce and even evaded attention in the past years. Although we strongly progressed in understanding its structural conformation with increasing detail, the field is lagging behind on the whereabouts of γ-secretase. This originates from its tetrameric stoichiometry and heterogeneity making it for a long time a challenging complex to study through more classic cell biological approaches. The advent of super-resolution microscopy techniques in recent years allows now to revisit this with an unprecedented resolution (*Schermelleh et al., 2019*). Here, we describe how PSEN1/γ-secretase complexes are organized and diffuse in the plane of the PM at single-molecule/complex resolution. We approached this through engineering PSEN and/or NCT deficient cells stably rescued with their fluorescently tagged versions allowing super-resolution analyses. The potential risk of overexpression artifacts is avoided through

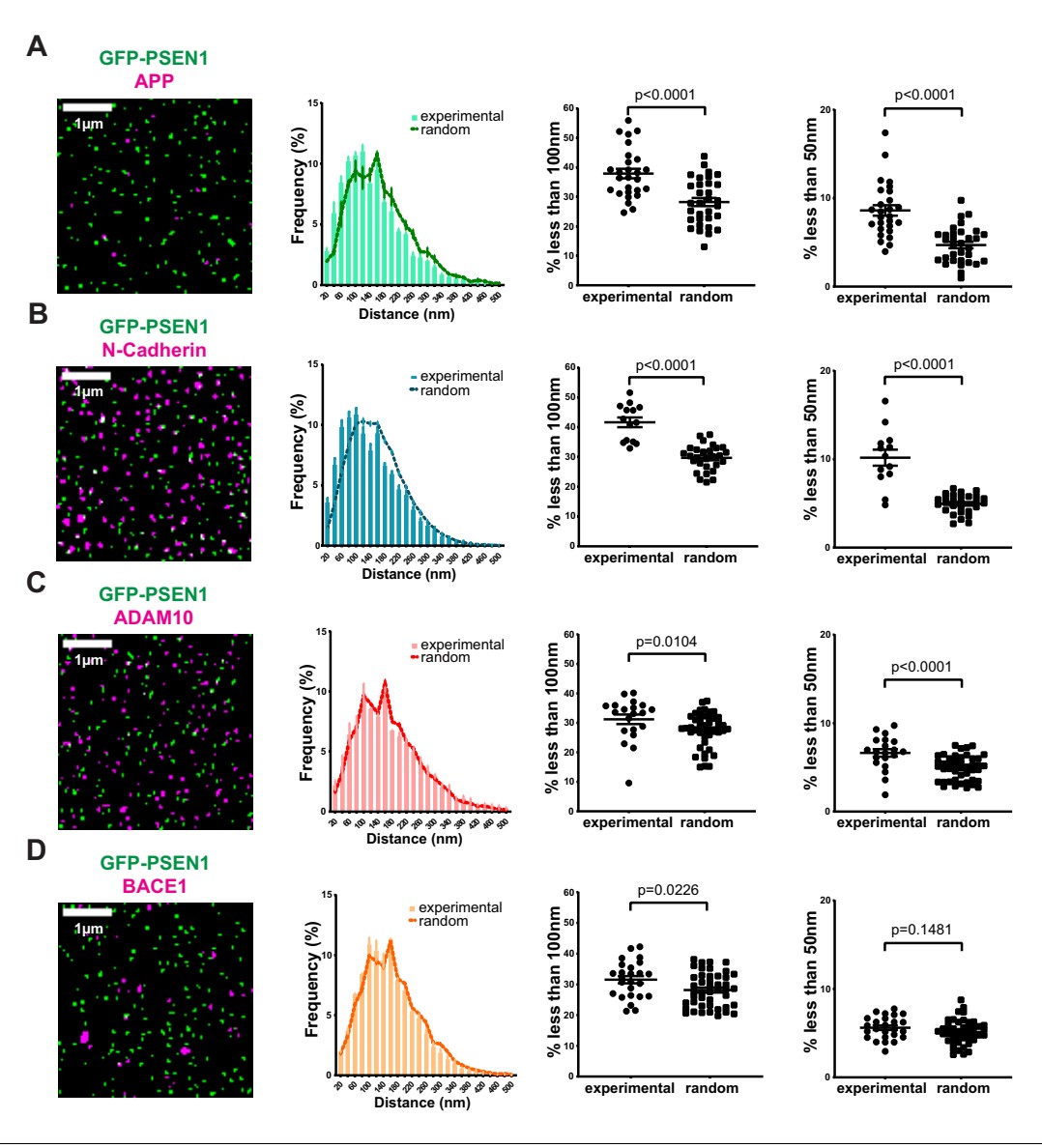

**Figure 4.** Correlation of PSEN1/γ-secretase complexes with substrates and sheddases at the cell surface. (From left to right) SIM mask, nearest-neighbor distances distribution, percentage of spots present at a distance <100 nm and percentage of spots present below 50 nm. Each dot represents one cell. (**A**) Distance of endogenous APP and (**B**) N-cadherin to GFP-PSEN1 shows a clear association to PSEN1 (mean ± SEM, n = 27 cells and n = 14 cells, respectively). (**C**) ADAM10-SNAP distance to GFP-PSEN1 shows a limited association to PSEN1 (mean ± SEM, n = 20 cells) whereas (**D**) BACE1-SNAP distance to GFP-PSEN1 shows no association to PSEN1 (mean ± SEM, n = 25 cells). All comparison analysis by two-tail Mann-Whitney test. See *Figure 4—source data 1*.
The online version of this article includes the following source data for figure 4:

**Source data 1.** Source Data for Nearest neighbor analysis.

the use of retro- or lentiviral vectors followed by FACS sorting to select pools of cells with similar physiological levels of expression of exogenously introduced γ-secretase subunits, as demonstrated by rigorous biochemical analysis. By starting from PSEN deficient cells, we could selectively focus on the dynamics of PSEN1 containing complexes that are known to distribute mainly at the cell surface and endosomal compartments (*Annaert et al., 1999*; *Kaether et al., 2002*; *Réchards et al., 2003*; *Sannerud et al., 2016*). Analysis of SIM-imaged PM sheets on our dual-tagged γ-secretase complex gave us quantitative evidence of subunit associations (*Figure 1*). Further experiments proved that

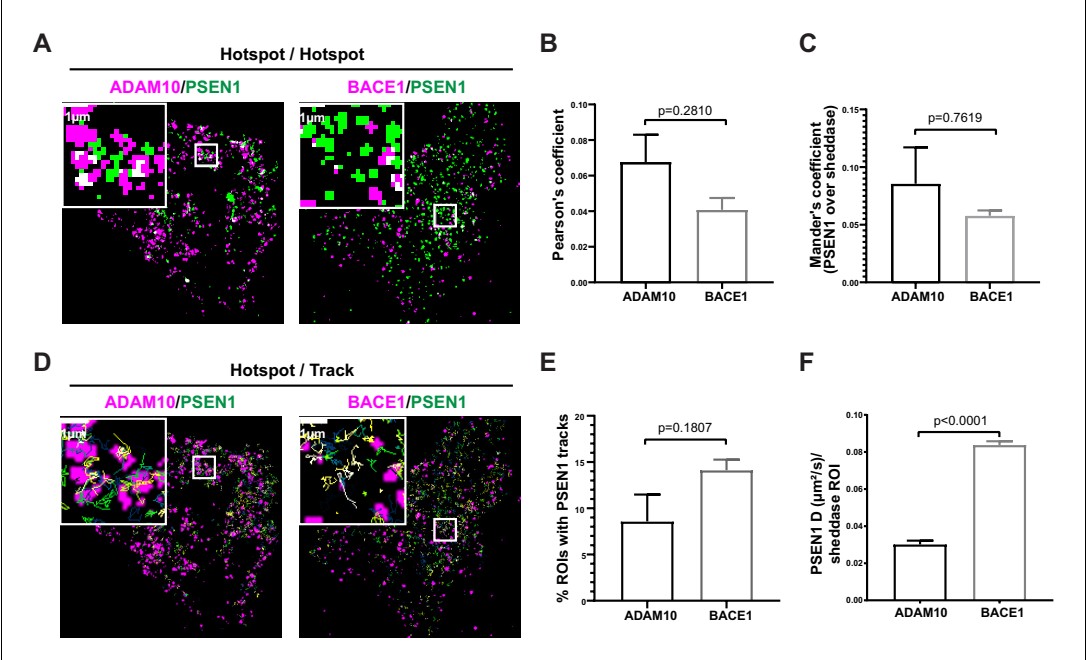

**Figure 5.** GSI-sensitive PSEN1/γ-secretase hotspots do not overlap with those of sheddases. (**A**) Diffraction-limited hotspots of SNAP tagged-sheddases and mEOS3.2-PSEN1 show limited overlap (white). Scale bar = 1 μm. (**B**) Pearson's coefficient of ADAM10 and BACE1 hotspots associated with PSEN1 hotspots show very low correlation. Comparison analysis by two-tail Mann-Whitney test (mean ± SEM, BACE1 n = 4 cells; ADAM10 n = 6 cells). (**C**) Mander's coefficient of PSEN1 hotspots over ADAM10 or BACE1 hotspot showed limited overlap. Comparison analysis by two-tail Mann-Whitney test (mean ± SEM, BACE1 n = 4 cells; ADAM10 n = 6 cells). (**D**) Tracks of mEOS3.2-PSEN1 show some association with diffraction-limited hotspots of SNAP tagged-sheddases. Scale bar = 1 μm. (**E**) Percentage of sheddases hotspots associated with mEOS3.2-PSEN1 tracks show a tendency for higher association with BACE1 hotspots. Comparison analysis by two-tail Mann-Whitney test (mean ± SEM, BACE1 n = 7 cells; ADAM10 n = 6 cells) (**F**) Diffusion coefficient analysis of pooled tracks associated with a sheddase-hotspot shows a significantly decreased diffusion coefficient for tracks on ADAM10-SNAP, but not for tracks on BACE1 hotspots. Comparison analysis by two-tail Mann-Whitney test (mean ± SEM, BACE1 = 263 tracks, n = 7 cells; ADAM10 = 126 tracks, n = 6 cells each).

these associations were in fact 1:1 molecular associations of NCT and PSEN1 in the same complex. To our knowledge, this is the first time that PSEN1/γ-secretase stoichiometry is demonstrated in situ.

Independent and complementary imaging strategies surprisingly demonstrate that PSEN1/γ-secretase laterally distributes at the PM as mono- and dimeric complexes (*Figure 2*). The latter is consistent with earlier biochemical evidence on the existence of dimeric complexes (*Hébert et al., 2003*; *Cervantes et al., 2004*) as well as in lower resolution structures (*Ogura et al., 2006*). When analyzed by blue native PAGE, mostly a ~ 440 kDa complex is reported possible coinciding with a dimeric configuration as the sum of the four components is about 210 kDa (*Kimberly et al., 2003*; *Fraering et al., 2004*; *Spasic et al., 2007*). A functional implication for monomers versus dimers is however not known, but could be potentially related to regions in the membrane with a higher density of complexes that promote dimerization. From a technical perspective, the fact that we can image PSEN1/γ-secretase as single complexes implicates that SIM analysis on PM sheets achieves single-molecule imaging, likely because imaging PM monolayers increases the signal to noise ratio.

Diffusion coefficients of membrane proteins range typically from ~0.0002 to 0.5 μm$^2$/s using single-particle tracking methods. SptPALM revealed very dynamic PSEN1/γ-secretase complexes with a mean diffusion coefficient of 0.15 μm$^2$/s, which is fast for a complex with twenty transmembrane domains (*Figure 3D*). Other multimembrane proteins like cystic fibrosis transmembrane conductance regulator (CFTR) diffuse extremely slow (0.005 μm$^2$/s) whereas G-protein coupled receptors, that also di-/oligomerize at the cell surface, have as well low mean diffusion coefficients (0.05–0.06 μm$^2$/s for GABA and $\beta_1/\beta_2$-adrenergic receptors [*Bates et al., 2006*; *Calebiro et al., 2013*]). Only rhodopsin and smoothened, both seven transmembrane domain proteins, have much higher coefficients (0.77 μm$^2$/s and 0.26 μm$^2$/s, respectively) (*Milenkovic et al., 2015*; *Kreutzberger et al., 2019*).

Interestingly, rhomboids, another member of the i-CLiP family, diffuse extremely fast with a mean diffusion coefficient of 0.88 $\mu m^2$/s (*Kreutzberger et al., 2019*). The mechanism behind this appears to be a hydrophobic mismatch with the surrounding lipids, causing a membrane distortion that reduces in turn membrane viscosity. As diffusion is the rate-limiting step in substrate catalysis, rhomboids overcame this by breaking the viscosity imposed 'speed limit' to facilitate substrate processing. PSEN1/γ-secretase has a tendency to be more mobile at the cell's edges, where it is assumed to be more active/required because of the higher presence and turnover of adhesion molecules (*Haapasalo and Kovacs, 2011*). Whether or not PSEN1/γ-secretase uses as well hydrophobic mismatch as a mechanism to diffuse across the PM needs to be further explored.

SptPALM revealed that PSEN1/γ-secretase changes motility during its diffusion, with frequent intervals of directed movement (*Figure 3C*). At the level of the PM, molecular motors are unlikely to act upon PM proteins (*Levi and Gratton, 2007*). Alternatively, directed motility has been explained by either the uniform flow or conveyor belt models (*Saxton, 1994*; *Saxton and Jacobson, 1997*). In the uniform flow model, particles in the membrane move together with the lipids at a constant velocity. In the conveyor belt model, the particle goes on and off from a structure that defines the directed motion, resulting in a mix of directed and other motilities. In support of the latter model, few studies suggested a functional interaction of PSEN1 with the actin cytoskeleton. For instance, PSEN1 has been reported to interact with filamin, an actin-binding protein (*Zhang et al., 1998*; *Guo et al., 2000*). More indirectly, absence of the orthologous PSEN in the moss *Physcomitrella patens*, causes cytoskeletal defects pointing to a scaffolding role of the moss PSEN between the cytoskeleton and adherent proteins (*Khandelwal et al., 2007*). To explore directed motility of PSEN1/γ-secretase in relation to the actin cytoskeleton, more appropriate cellular models including migrating cells, or the neuronal growth cone might be considered.

Different types of motilities have been correlated to transient protein-protein associations, changes in lipid microenvironment and/or changes in protein conformational state (*Suzuki et al., 2012*; *Sibarita, 2014*; *Constals et al., 2015*; *Freeman et al., 2018*). Anchoring to the cytoskeleton usually results in an immobile state as shown for CD44 and integrin β₁ and β₂ (*Rossier et al., 2012*; *Freeman et al., 2018*). Confined motility of α₂-macroglobulin receptor (α2MR) and transferrin receptor (TfR) has been related to the enclosure of the protein in between actin filaments of the cytoskeleton, leading to the 'picket-fence' hypothesis (*Kusumi et al., 1993*; *Sako and Kusumi, 1994*). In the AMPAR, conformational changes triggered by binding of glutamate leads to a desensitized state correspondent to a switch from immobile to mobile molecules (*Constals et al., 2015*). Although experimental and simulated data show that PSEN1/γ-secretase also undergoes conformational changes upon substrate binding or in the presence of inhibitors (*Bai et al., 2015a*; *Aguayo-Ortiz and Dominguez, 2018*; *Yang et al., 2019*; *Zhou et al., 2019*), sptPALM did not reveal changes on diffusion coefficients, distribution of tracks or types of motility following treatment with GSIs. Both GSIs only showed a trend in decreased motility of PSEN1/γ-secretase in the edges of the cell. Further analysis of sptPALM localizations with DBSCAN revealed hotspots with a higher density of PSEN1/γ-secretase tracks, which became reduced in the presence of GSIs (*Figure 3L*). We currently do not know the nature of these hotspots, but they could relate to specific nano- or microdomains that favor the presence of PSEN1/γ-secretase because of a specific lipid microenvironment, or underlying structures like cytoskeletal elements. PSEN1/γ-secretase could be recruited to these hotspots when in a specific activation state and/or conformation. If these hotspots were sites of substrate cleavage, locking the conformation in the active or transition state would mimic this causing an increase in the number or size of such hotspots. However, the opposite is seen, indicating that inactive PSEN1/γ-secretase complexes might settle down here. Support also comes from the observation that PSEN1/γ-secretase hotspots did not associate with those of the sheddases ADAM10 and BACE1, which would be preferential locations where CTF substrates for PSEN1/γ-secretase would be generated. Whether PSEN1/γ-secretase dynamic hotspots are indeed favored by inactive PSEN1/γ-secretases, requires additional experimentation using, for instance, inactive aspartate mutants.

When analyzing individual tracks of PSEN1/γ-secretase, a small portion of tracks frequented hotspots of both ADAM10 and BACE1. However, while they appeared to pass by in the case of BACE1 hotspots, the motility, as measured by mean diffusion coefficient, decreased in the case of ADAM10 to ~0.04 $\mu m^2$/s (*Figure 5F*). This is in line with the association found between PSEN1/γ-secretase and ADAM10-SNAP using nearest-neighbor analysis (*Figure 4*). This is not detected for BACE1, which may not be that surprising given that BACE1 activity is more favorable in endosomes

(*Sannerud et al., 2011*; *Das et al., 2016*) and that the potential encountering of BACE1 via the produced substrates with PSEN1/γ-secretase complexes would more likely to be found there (*Rajendran et al., 2006*; *Wang, 2018*). Nevertheless, as compared to the substrate APP, this ADAM10 association is very moderate, also considering, the density of ADAM10-SNAP (three spots/$\mu m^2$) compared to the even lower density of APP (0.5 spots/$\mu m^2$). Together with the largely mono- and dimeric nature of PSEN1/γ-secretase at the PM, this questions the existence of preformed megadalton-sized complexes, which suggests multiple copies of either ADAM10 or BACE1 and PSEN1/γ-secretase (*Chen et al., 2015*; *Liu et al., 2019*). Furthermore, to our knowledge, sheddases have not been identified in the several interactome studies using PSENs as a bait (*Wakabayashi et al., 2009*; *Jeon et al., 2013*), whereas this was the case for the substrate APP-C83 (*Esler et al., 2002*). The support for the megadalton-sized complexes concept is majorly biochemical in nature and based on the void fractions of gel filtration-based separations of crude detergent extracts. Thus, it cannot be excluded that such macrocomplexes may be formed during or after extraction. Our single-particle analysis rather favors the idea of a 1:1 encountering of PSEN1/γ-secretase to areas where sheddases reside to 'accept' the respective substrate CTF for further processing. At most, this would result in short temporal or transient associations via substrate transfer.

In conclusion, the presented single-molecule resolution analyses uncovered the distribution and dynamics of PSEN1/γ-secretase at the PM, as well as show, for the first time, the stoichiometry of the complex while embedded in its natural environment. These findings will pave the way for studying the association of γ-secretase with substrates or regulatory proteins in real-time. Ultimately, the spatial and temporal regulation of consecutive processing by a sheddase and γ-secretase could be obtained. We anticipate that single-molecule resolution approaches will become a standard in the field to further decipher the dynamics of intramembrane proteolysis particularly in the neuron. Herein, the distribution and dynamics of these enzymes at the growth cone and synapse will be of particular interest given that many γ-secretase substrates are adhesion proteins functioning in axonal outgrowth and synapse formation and stability (*Jurisch-Yaksi et al., 2013*). Whereas their neuronal functions are slowly unveiled, indeed little is known about the dynamics of regulated intramembrane proteolysis in the synapse, let alone how for instance Familial Alzheimer's Disease (FAD)-associated mutations in PSEN1 or APP alter these dynamics and thus the functional readouts of the synapse. In an Alzheimer's disease context, synaptic plasticity is affected in absence of PSENs and in PSEN1 FAD models (*Saura et al., 2004*; *Priller et al., 2007*). Hence, characterizing the distribution, associations and dynamics of γ-secretase in the synapse would give a much more detailed insight on the molecular mechanisms driving the disease.

# Materials and methods

## Key resources table

| Reagent type (species) or resource | Designation | Source or reference | Identifiers | Additional information |
|---|---|---|---|---|
| Strain, strain background (*Escherichia coli*) | XL10-Gold | Stratagene | 200314 | Competent cells |
| Cell line (*M. musculus*) | MEF NCT/PSEN1/PSEN2 TKO NCT-SNAP GFP-PSEN1 | This paper | | Mouse embryonic fibroblast KO of NCT, PSEN1 and PSEN2, transduced to express mNCT-SNAP and GFP-hPSEN1 |
| Cell line (*M. musculus*) | MEF PSEN1 KO mCherry-PSEN1 | *Sannerud et al., 2016* | | Mouse embryonic fibroblast KO of PSEN1, transduced to express mCherry-hPSEN1 |
| Cell line (*M. musculus*) | MEF PSEN1 KO GFP-PSEN1 | *Sannerud et al., 2016* | | Mouse embryonic fibroblast KO of PSEN1, transduced to express GFP-hPSEN1 |
| Cell line (*M. musculus*) | MEF PSEN1/PSEN2 KO GFP-PSEN1 | *Sannerud et al., 2016* | | Mouse embryonic fibroblast KO of PSEN1 and PSEN2, transduced to express GFP-hPSEN1 |

*Continued on next page*

*Continued*

| Reagent type (species) or resource | Designation | Source or reference | Identifiers | Additional information |
|---|---|---|---|---|
| Cell line (*M. musculus*) | MEF NCT KO NCT-SNAP | This paper | | Mouse embryonic fibroblast KO of NCT, transduced to express mNCT-SNAP |
| Cell line (*M. musculus*) | MEF PSEN1 KO GFP-PSEN1 + BACE1-SNAP | This paper | | Mouse embryonic fibroblast KO of PSEN1, transduced to express GFP-PSEN1 and hBACE1-SNAP |
| Transfected construct (*H. sapiens*) | GFP-hPSEN1 | This paper | | Construct to transduce MEF PSEN1 KO, tKO and dKO |
| Transfected construct (*H. sapiens*) | mEOS3.2-hPSEN1 | This paper | | Construct to transduce MEF PSEN1 KO, tKO and dKO |
| Transfected construct (*H. sapiens*) | mCherry-hPSEN1 | This paper | | Construct to transfect and express mADAM10-SNAP |
| Transfected construct (*H. sapiens*) | mNCT-SNAP | This paper | | Construct to transduce tKO and NCT KO |
| Transfected construct (*H. sapiens*) | mNCT-GFP | This paper | | Construct to transduce NCT KO |
| Transfected construct (*H. sapiens*) | mNCT-YFP | This paper | | Construct to transduce NCT KO |
| Transfected construct (*H. sapiens*) | hBace1-SNAP | This paper | | Construct to transduce PSEN1 KO GFP-PSEN1 and mEOS3.2-PSEN1 |
| Transfected construct (*H. sapiens*) | mADAM10-SNAP | This paper | | Construct to transfect and express mADAM10-SNAP |
| Transfected construct (*H. sapiens*) | pX330-PSEN1 | This paper | | CRISPR plasmid to KO PSEN1 |
| Transfected construct (*H. sapiens*) | pX330-PSEN2 | This paper | | CRISPR plasmid to KO PSEN2 |
| Transfected construct (*H. sapiens*) | px459-NCT | This paper | | CRISPR plasmid to KO NCT |
| Antibody | anti-LAMP1 (rat monoclonal) | Santa Cruz | sc-19992; RRID:AB_2134495 | (1:200) |
| Antibody | anti-NCT (mouse monoclonal) | *Esselens et al., 2004* | 9C3 | (1:7000) |
| Antibody | anti-PSEN1-NTF (rabbit polyclonal) | Abcam | ab71181; RRID:AB_1603935 | (1:2000) |
| Antibody | anti-PSEN1-NTF (rat polyclonal) | Millipore | MAB1563; RRID:AB_1671560 | (1:4000) |
| Antibody | anti-PSEN1-CTF (rabbit monoclonal) | Abcam | ab76083; RRID:AB_1310605 | (1:2000) |
| Antibody | anti-PEN2 (rabbit polyclonal) | Abcam | ab18189; RRID:AB_444310 | (1:1000) |
| Antibody | anti-APP-CTF (rabbit polyclonal) | *Esselens et al., 2004* | B63 | (1:10,000) |
| Antibody | anti-transferrin receptor (mouse monoclonal) | Invitrogen | 136800; RRID:AB_2533029 | (1:4000) |

*Continued on next page*

Continued

| Reagent type (species) or resource | Designation | Source or reference | Identifiers | Additional information |
|---|---|---|---|---|
| Antibody | anti-rabbit (HRP-goat) | Bio-Rad | 1706515; RRID:AB_11125142 | (1:10000) |
| Antibody | anti-mouse (HRP-goat) | Bio-Rad | 1706516; RRID:AB_11125547 | (1:10000) |
| Antibody | anti-GFP (rabbit polyclonal) | Bio-Rad | A11122; RRID:AB_221569 | (1:10000) |
| Antibody | anti-PSEN1-CTF (mouse monoclonal) | Millipore | MAB5232; RRID:AB_95175 | (1:1000) |
| Antibody | anti-mouse Alexa Fluor790 (goat) | Invitrogen | A11375; RRID:AB_2534146 | (1:15,000) |
| Recombinant DNA reagent | pSNAPf (plasmid) | New England Biolabs | N9183S | SNAPtag sequence |
| Peptide, recombinant protein | γ-secretase | *Szaruga et al., 2017* | | |
| Commercial assay or kit | NEBuilder HiFI DNA assembly mix | New England Biolabs | E5520 | Gibson assembly |
| Commercial assay or kit | Q5 site-directed mutagenesis | New England Biolabs | E0554S | Mutagenesis |
| Commercial assay or kit | NativePAGE | ThermoFisher | | Native protein electrophoresis |
| Commercial assay or kit | MycoAlert | Lonza | LT07-118 | Mycoplasma detection kit |
| Commercial assay or kit | DAPT | Tocris Bioscience | 2634/10 | γ-secretase inhibitor |
| Commercial assay or kit | Inhibitor X | Calbiochem | 565771 | γ-secretase inhibitor |
| Commercial assay or kit | DMSO | VWR | A3672 0250 | Treatment vector |
| Software, algorithm | BD FACS | *Stall and AC Technologies, 2008* BD Biosciences | RRID:SCR_005400 | https://www.bdbiosciences.com/en-us/instruments/research-instruments/research-software/flow-cytometry-acquisition/facsdiva-software |
| Software, algorithm | QuASIMoDOH | *Paparelli et al., 2016* | | Software to analyze spots distribution |
| Software, algorithm | Nearest Neighbour analysis | MATLAB | RRID:SCR_001622 | findNearestNeighbors (ptCloud, point, K) |
| Software, algorithm | B-unwarpJ | *Schindelin et al., 2012* ImageJ | | https://imagej.net/BUnwarpJ |
| Software, algorithm | H-watershed | *Schindelin et al., 2012* ImageJ | | https://imagej.net/Interactive_Watershed |
| Software, algorithm | Spot intensity analysis | *Schindelin et al., 2012* ImageJ | | https://imagej.net/Spot_Intensity_Analysis |
| Software, algorithm | Molecular counting by photon statistics | *Grußmayer and Herten, 2017* PicoQuant | | |
| Software, algorithm | qSR | *Andrews, 2017* | | https://github.com/cisselab/qSR |
| Software, algorithm | PALMtracer | University of Bordeaux | | https://www.iins.u-bordeaux.fr/team-sibarita-PALMTracer |
| Software, algorithm | Thunderstorm | *Ovesný et al., 2014* | RRID:SCR_016897 | https://zitmen.github.io/thunderstorm/ |
| Software, algorithm | SR Tesseler | *Levet et al., 2015* | | https://www.iins.u-bordeaux.fr/team-sibarita-sr-tesseler |

*Continued on next page*

*Continued*

| Reagent type (species) or resource | Designation | Source or reference | Identifiers | Additional information |
|---|---|---|---|---|
| Other | SNAP-Cell 647-SiR substrate | New England Biolabs | S9102S | Cell permeable substrate to label SNAPtag |
| Other | Tetraspeck beads | Invitrogen | S9102S | Fiducial Markers (1:1000) |

## Antibodies

For confocal microscopy, we used the rat monoclonal anti-LAMP1 (sc-19992, Santa Cruz, 1:200). For WB, the following antibodies were used: mouse monoclonal anti-NCT (*Esselens et al., 2004*) (9C3, 1:7000), rabbit polyclonal anti-PSEN1-NTF (ab71181, Abcam, 1:2000) or rat polyclonal anti-PSEN1-NTF for co-immunoprecipitation (MAB1563, Millipore, 1:4000), rabbit monoclonal anti-PSEN1-CTF (ab76083, Abcam, 1:2000), rabbit polyclonal anti-PEN2 (ab18189, Abcam, 1:1000), rabbit polyclonal anti-APP-CTF (*Esselens et al., 2004*) (B63, 1:10,000) and mouse monoclonal anti-transferrin receptor (136800, Invitrogen, 1:4000). Secondary antibodies included HRP-conjugated goat-anti-mouse and goat-anti-rabbit (Bio-Rad, 1:10000 dilution). Co-immunoprecipitation was done with rabbit anti-GFP (#A11122; Invitrogen). For quantitative WB, mouse monoclonal anti-PSEN1-CTF (MAB5232, Millipore, 1:1000 dilution) and near-infrared goat anti-mouse Alexa Fluor790 (#A11375, Invitrogen, 1:15,000 dilution) were used. For PM sheet immunofluorescence, GFPbooster-Atto647N (Chromotek, 1:1000 dilution) was used.

## Expression vectors

Unless stated otherwise, cloning was performed using Gibson assembly (*Gibson et al., 2009*) with the NEBuilder HiFi DNA assembly mix (New England Biolabs). For NCT-GFP and NCT-SNAP, the cDNA encoding GFP was amplified from pEGFP-N1 using primers 5'-gcggccgcctgcaggtcgacaatgcggccgctggaggtggaggtagtggaggtggaggttcagtgagcaagggcgaggagctgttcacc-3' and 5'-agtcggatccagtcgtcgacctacttgtacagctcgtccatgcc-3' and SNAPtag was amplified from pSNAPf (New England Biolabs), with 5'-gcggccgcctgcaggtcgacaatgcggccgctggaggtggaggtagtggaggtggaggttcagacaaagactgcgaaatgaagcg-3' and 5'-agtcggatccagtcgtcgacctaacccagcccaggctt-3' primers. Each product was inserted in the pMSCV-NCT plasmid (*Chávez-Gutiérrez et al., 2008*), containing the WT NCT mouse sequence following digestion with *SalI* and alkaline phosphatase treatment. The stop codon was deleted by Q5 site-directed mutagenesis (New England Biolabs) using the primers 5'-aatgcggccgctggaggt-3' and 5'-gtaagacacagctcctggctctc-3'. For both constructs, the puromycin resistance cassette was exchanged for a hygromycin cassette. Hereto, vectors were linearized by PCR using the primers 5'-ggaatagcgcccgccccacg-3' and 5'-gcttttttcatggtaagcttgggctgc-3', removing the puromycin cassette. The hygromycin cassette was amplified from pCHMWS-GFP-ires-hygro plasmid by the primer 5'-caagcttaccatgaaaaagcctgaactcaccg-3' and 5'-ggcgggcgctattcctttgccctcggacg-3' and fused to the linearized vectors. For NCT-YFP, the pGEMt-NCT plasmid containing WT NCT mouse sequence was digested at *EcoRI/SalI* sites. A spacer containing *EcoRI/SalI* sites was amplified with the primers 5'-GGAATTCCAACAGGACAG-3' and 5'-tcggtcgacatagtttagcggccgccagatccgccagatccgccagatccgccgtaagacacagctcctgg-3' and ligated with the linearized vector using T4 ligase. The pGEMt-NCT-spacer construct was digested at *NotI/SalI* sites. YFP was amplified from pEYFP-N1 using the primers 5'-ataagaatgcggccgcatggtgagcaagggcgag-3' and 5'-tcggtcgacctattacttgtacagctcgtc-3', which included *NotI* and *SalI* sites, respectively. The product was digested with *NotI/SalI* and ligated into the pGEMt-NCT-spacer. The construct pGEMt-NCT-spacer-YFP (NCT-YFP) was then moved to a pMSCV* retroviral vector by restriction cloning at *NcoI/SalI* sites. For mEOS3.2-PSEN1, the pMSCV*-GFP-hPSEN1 construct (*Sannerud et al., 2016*) was linearized by *BglII* digestion and mEOS3.2 was amplified from pmEOS3.2-N1 with the primers 5'-taggcgccggaattagatctatgagtgcgattaagccagacatg-3' and 5'-cgaagcttgagctcgagatcttcgtctggcattgtcaggcaat-3' containing a nine amino acid linker between mEOS3.2 and PSEN1. Fragments were ligated by Gibson assembly. For pCDH(zeo)-BACE1-SNAP, the puromycin resistance cassette of pCDH-wt (*Boncompain et al., 2012*) was exchanged for the zeocin cassette from pCDNA(zeo) (ThermoFisher). The linearized fragments, primers 5'-aggactgagtcgacaatcaacctctgga-3' and 5'-ttggccatctagcgtaggcgccg-3' for the vector and 5'-acgctagatggccaagttgaccagtgc-3' and 5'-attgtgactcagtcctgctcctcggc-3' for the insert, were fused. Human BACE1 was amplified from pcDNA-hBACE1 (gift from C. Haass. Ludwig-

Maximilians-Universität, Adolf-Butenandt-Institute, Munich, Germany) using primers 5'-gttttgacctcca-tagaagattctagaatggcccaagccctgc-3' and 5'-tctttgtctgaaccgcctccacccttcagcagggagatg-3'; as well as, SNAP tag by the primers 5'-gcggttcagacaaagactgcgaaatgaagcg-3' and 5'-gatcgcagatccttgcggccgcc-taacccagcccaggctt-3'. pCDH(zeo) vector was linearized by restriction using *XbaI* and *NotI* sites. The three fragments were fused. For pCDNA(zeo)-mADAM10-SNAP (*Tousseyn et al., 2009*), we linearized pCDNA(zeo)-mADAM10 by restriction using *KpnI* site. SNAP tag fragment was amplified using the primers 5'- atccgagctcggtacgacaaagactgcgaaatgaagcg-3' and 5'- ggtttaaacttaagcttggtacttaat-taacctcgagtttaaacgcggatcc-3', and fragments were fused. To remove the STOP codon, we used Q5 site-directed mutagenesis with primers 5'-atgcgacgctcaggatccgag-3' and 5'- gtgtcccatttga-taactctctc-3'. All constructs were sequenced.

## Generation of stable cell lines

MEFs deficient in either PSEN1 or both PSEN1 and −2 (*Herreman et al., 1999*; *Herreman et al., 2000*; *Nyabi et al., 2002*) were stably rescued using retro- or lentiviral particles. GFP-PSEN1 rescued dKO and GFP- and mCherry-PSEN1 rescued sKO MEFs have previously been described (*Sannerud et al., 2016*). In addition, rescue lines with mEOS3.2-hPSEN1 were generated. Briefly, KO MEFs were exposed to different amounts of virus loads and selected by puromycin resistance (3 µg/ml). Stable pools were validated by WB and confocal microscopy to confirm γ-secretase maturation, activity, and localization. For PSEN1/PSEN2/NCT tKO MEFs, we used the web-based CRISPR Design Tool (http://crispor.tefor.net/) to select the genomic sequence target in mouse PSEN1 (5'-caacgttat-caagtacctccccgaa-3'), PSEN2 (5'-caacgtcctgggcgaccgtcgggcc-3') and NCT (5'-caccgcgttgagcaggcg-gacaca-3'). Oligo pairs (Integrated DNA Technologies) encoding guide sequences were annealed and ligated into the plasmid pX330 (Addgene) for PSEN1 and PSEN2 and px459 for NCT following Zhang's laboratory protocol (https://www.addgene.org/crispr/zhang/). To generate PSEN1 and 2 dKO, MEFs were co-transfected with pX330-PSEN1 and pX330-PSEN2 using FugeneHD (Promega). The selection of dKO clones was done by serial dilution followed by western blot analysis. Once a PSEN1 and 2 dKO clone was selected, cells were transfected with the px459-NCT vector followed by three days of selection with puromycin to select for transfected cells. Cells were amplified and selection of independent NCT KO clones was done by serial dilution followed by WB analysis. The rescue of tKO MEFs was done in two steps. First, retroviral mNCT-SNAP or mNCT-GFP particles transduced tKO cells and selected with hygromycin (100 µl/ml). Selected cells were transduced again with either GFP- or mEOS3.2-hPSEN1 retroviral particles and selected with puromycin. Double transduced pools were FAC Sorted into four populations with different levels of each protein and checked by confocal and WB. We assessed the stoichiometry and integrity of the complex by blue native gel electrophoresis and co-immunoprecipitations on SPION-isolated PMs. For NCT KO, the same CRISPR strategy with the aforementioned NCT sequence was used. Rescue with NCT-SNAP was done in the same fashion as with tKO. For NCT KO NCT-SNAP/NCT-GFP, selected NCT KO NCT-SNAP MEFs were again transduced with mNCT-GFP and selected with puromycin. For cells expressing sheddases, we transduced sKO GFP-PSEN1 and sKO mEOS3.2-PSEN1 MEFs with hBACE1-SNAP lentiviral particles and selected with zeocin (200 µg/ml). Selected cells were then FAC Sorted to select low expression cells. For mADAM10-SNAP, sKO GFP-PSEN1 and sKO mEOS3.2-PSEN1 MEFs were transiently transfected with FugeneHD (Promega), following manufacturer's recommendations. All cell lines were evaluated by WB and confocal microscopy. Cell lines were screened for mycoplasma yearly using MycoAlert mycoplasma detection kit (Lonza) and were authenticated by STR profiling (Microsynth AG).

For SNAP-tag labeling, we used the SNAP-Cell 647-SiR substrate (New England Biolabs) diluted in DMSO. About 1 hr prior to cell processing, cells were incubated with 0.15 µM SiR-SNAP in complete medium for 15 min at 37°C. Cells were washed three times with complete medium and incubated with fresh medium for 30 min to 1 hr at 37°C, after which cells could be further processed. Finally, and following labeling, cells were trypsinized and stained with DAPI for dead cell exclusion. Around $3*10^6$ cells were sorted (BD FACSAria III; BD Biosciences), using the FACSDiva software or sterile sorted (BD Influx; BD Biosciences), using the BD FACS software. Four populations were defined according to the relative abundance of NCT-SiR and GFP- or mEOS3.2- (green) PSEN1: P5 (NCT low/PSEN1 low), P6 (NCT high/PSEN1 low), P7 (NCT low/PSEN1 high) and P8 (NCT high/PSEN1 high). The different populations were collected for subsequent culture and further validated using western blot analysis and confocal microscopy.

## Purification of γ-secretase

Protease complexes were expressed in Hi5 insect cells using recombinant baculoviruses bearing the WT subunits of the human protease complex (NCT-GFP, PSEN1, APH1 and PEN2), as purified as described in *Szaruga et al., 2017*. Briefly, total membrane fraction was prepared from Hi5 cells overexpressing the protease complex and membrane proteins extracted in 2% CHAPSO buffer (25 mM Pipes pH 7.4, 300 mM NaCl, 5% Glycerol, PI). γ-Secretase complexes were purified using an anti-GFP nanobody covalently coupled to agarose beads (NHS-activated beads, GE Healthcare) and untagged γ-complexes eluted by proteolysis (PreScission protease cleavage site present between NCT and GFP).

## Gel electrophoresis and western blot

For the preparation of microsomal membranes, cells were grown in three 10 cm dishes to 90% confluence, scraped in ice-cold PBS$^{-/-}$ and spun down (4 min,1500xg, 4°C). Pellets were resuspended in homogenization buffer (5 mM Tris-HCl, 250 mM sucrose, 1 mM EGTA, 1x cOmplete protease inhibitor (Roche), pH 7.4) and homogenized by 10 cycles through a cell cracker (10 μm clearance ball). After centrifugation (10 min, 800 g, 4°C), post-nuclear supernatant was further centrifuged (1 hr, 126203xg, 4°C). Microsomal membranes were resuspended in 25BTH20G (25 mM BisTris-HCl, 20% glycerol, pH 7.0) with 0.5% n-Dodecyl β-D-maltoside (DDM; Sigma) and incubated 4 hr at 4°C. Samples were cleared by two subsequent centrifugation steps (30 min and 15 min, 126203xg, 4°C). Both supernatant fractions were mixed and protein concentration was determined by the Bio-Rad DC protein assay (Bio-Rad). Blue native sample buffer (5x; 2.5% Coomassie Blue G-250 (Pierce), 50 mM Bis Tris-HCl, 15% sucrose, 250 mM 6-aminocaproic acid, pH 7.0) was added to 10 μg of the sample. BN-PAGE was performed at 4°C in a 4–16% Native PAGE polyacrylamide gel (Invitrogen). The high molecular weight marker (Amersham) was prepared according to the manufacturer's instructions. Electrophoresis was done at 100V for 20 min, followed by 30 min at 150V in anode buffer (1x Native-PAGE running buffer; Invitrogen) and dark blue cathode buffer (1x NativePAGE running buffer, 1x NativePAGE Cathode Additive; Invitrogen). Subsequently, the cathode buffer was replaced with a light blue cathode buffer (1x NativePAGE running buffer, 0.1xNativePAGE Cathode Additive; Invitrogen) and run for 3 hr at 200V. The gel was washed with 0.1% SDS and 10% methanol in 1x Bolt transfer buffer (Invitrogen) and transferred to an Immobilon-P PVDF membrane (Millipore). The blot was unstained for 1 hr in 30% MeOH and 10% acetic acid and processed for western blotting. For regular western Blotting, cells were cultured to ~80% confluency, lysed and protein concentration determined using the Bio-Rad DC protein assay (Bio-Rad). Samples were subjected to SDS-PAGE using precast Bolt 4–12% Bis-Tris plus gels (Invitrogen) followed by western blotting. Membranes were blocked for 1 hr and probed with primary antibodies overnight at 4°C followed by incubation with secondary antibodies for 1 hr at room temperature. Membranes were developed using the enhanced chemiluminescence kit (Western Lightning-Plus ECL, PerkinElmer), imaged on the Fuji Min-iLAS 3000 imager (Fuji, Düsseldorf, Germany) and analyzed using Aida Image Analyzer software (Raytest, Germany). For quantitative western blotting, 1 μg of the PM fraction and 33 ng of recombinant γ-secretase were run and transferred using the NuPAGE system (Invitrogen). Membranes were blocked for 1 hr with 1% BSA and probed with primary antibodies overnight at 4°C. Next, near-infrared secondary antibodies were probed for 1 hr at RT followed by quantitative analysis (Amersham Typhoon) using ImageQuant (Amersham). For quantification, recombinant γ-secretase was used as a standard.

## Isolation of plasma membranes
### Biochemical

PM fractions are isolated as previously reported (*Tharkeshwar et al., 2017*). Briefly, cells were grown in 10 cm dishes to ~95% confluence. Then, incubated on ice for 30 min, washed with PBS$^{-/-}$ and, incubated with 1 mg/dish of aminolipid-coated superparamagnetic iron oxide nanoparticles (SPIONs) for 30 min on ice. Cells were washed, collected by scraping and pelleted by centrifuging (10 min, 200xg, 4°C). The pellet was resuspended in homogenization buffer (200 mM sucrose/10 mM HEPES/1 mM EDTA/1 x cOmplete protease inhibitor (Roche), pH7.4) and homogenized by cell cracking (10 μm clearance ball). The total cell lysate was centrifuged (10 min, 200xg, 4°C) and the post-nuclear supernatant was passed through a magnetic LS column (Miltenyi Biotec) held in a

magnetic field (SupermacsII magnet) to retain SPION loaded PMs. The column was washed with homogenization buffer, 1M KCl and 0.1M $Na_2CO_3$, and eluted in 0.1M $Na_2CO_3$ buffer. The eluted sample was centrifuged (1 hr, 125,000xg$_{max}$, 4°C) and resuspended in 0.1M $Na_2CO_3$. Protein concentration was determined as described above. For subsequent co-immunoprecipitation, PM fractions were centrifuged (1 hr, 125,000xg$_{max}$, 4°C) to exchange buffer. The pellet was resuspended in 1% CHAPSO buffer (50 mM HEPES/120 mM NaCl/1% CHAPSO/1 x cOmplete protease inhibitor EDTA free) and incubated for 20 min on ice. The lysate was collected after centrifugation (15 min, 14,000xg$_{max}$, 4°C). Immunoprecipitation of ~5 μg total protein with anti-GFP was done for 16 hr at 4°C with end-over-end rotation. The sample was then incubated 1 hr at 4°C with Protein G agarose (Pierce). Beads were spinned-down by centrifuging (1 min, 1500xg, 4°C). The supernatant (unbound fraction) was collected and protein was precipitated as previously described (*Wessel and Flügge, 1984*). Briefly, the sample was vortexed with 40%MeOH and 20% Chloroform for 2 min. After centrifugation (5 min, 20800xg, 4°C), the upper phase was discarded, fresh MeOH was added and the sample vortexed for 2 min. After 5 min centrifugation, the supernatant was discarded and the pellet was left to air dry. For the bound fraction, after washing the beads four times using 1% CHAPSO buffer, bound proteins were eluted by incubation of beads in 4x LDS sample buffer (Invitrogen) with 4% β-mercaptoethanol for 10 min at 70°C and subjected to SDS-PAGE and western blotting.

## Imaging

PM sheets were prepared as described (*Paparelli et al., 2016*) with minor modifications. In brief, cells were grown on 1.5 mm coverslips to 80% confluence. Coverslips were transferred to ice, washed twice with ice-cold PBS$^{+/+}$ and twice with coating buffer (20 mM MES, 135 mM NaCl, 0.5 mM $CaCl_2$, 1 mM $MgCl_2$, pH5.5) and incubated with 1% silica beads in coating buffer for 30 min on ice. After rinsing with deionized water (10 min), a shear force was applied using a syringe held at a 30° angle, followed by thorough washing. Supported PM sheets were fixed in 4% formaldehyde for 1 hr on ice, rinsed and incubated for 10 min with Tetraspeck beads (Invitrogen, 1:1000) in a humid chamber. Then, coverslips were rinsed three times in ddH$_2$O and mounted with Mowiol. If the SNAP-tag was present in the cells, covalent binding of the SiR dye to SNAP was performed in living cells prior to PM sheet preparation. For GFPbooster staining of PM sheets, after fixation, coverslips were transferred to a humid chamber and blocked (PBS containing 2% BSA, 2% FCS, 0.2% fish gelatin, 5% goat serum) for 1 hr. Incubation with GFPbooster-Atto647N in blocking buffer was done for 1 hr at RT. After 1 hr rinsing in PBS$^{-/-}$, coverslips were incubated with Tetraspeck beads (1:1000, Invitrogen), rinsed three times in ddH2O and mounted with Mowiol. As a quality control, PM sheets were fixed and analyzed by scanning EM.

## Imaging and image analyses

### Scanning electron microscopy

PM sheets were prepared on coverslips, and fixed with 2.5% glutaraldehyde (Polysciences, Inc) in 0.1 M sodium cacodylate buffer (16 hr, 4°C). Samples were washed in cacodylate buffer and post-fixed with 1% osmium tetraoxide (1 hr on ice). After three washes of 5 min in water, samples were gradually dehydrated by 5 min incubation steps in 50%, 70%, 96% and three times 100% ethanol. Samples were then critical point dried (Leica CPD300), mounted on a pin stub and sputter-coated with 4 nm chromium (Leica ACE600). Imaging was done with secondary- and back-scatter electron detectors in a Zeiss Sigma scanning electron microscope at an accelerating voltage of 5kV.

### Confocal microscopy

Cells were grown to 70% confluency on coverslips and stained with SiR-substrate as suggested by the producer (New England Biolabs). Cells were washed twice with Dulbecco's PBS$^{+/+}$ (Invitrogen) at RT and fixed for 30 min in 4% formaldehyde (Sigma) supplemented with 4% sucrose in 120 mM sodium phosphate buffer, pH7.3. Cells were permeabilized with 0.1% triton, blocked and incubated with primary antibodies, followed by staining with Alexa-dye coupled secondary antibodies (Invitrogen). Coverslips were rinsed three times in ddH$_2$O and mounted with Mowiol (Sigma). Cells were imaged with a Nikon C2 Eclipse Ni-E inverted confocal equipped with an oil-immersion plan Apo 60 × 1.4 NA objective lens. Images were processed with ImageJ.

## Structured Illumination Microscopy (SIM) and QuASIMoDOH analysis

PM sheets with 100 nm Tetraspeck beads (Invitrogen) were imaged by widefield microscopy and SIM on a Zeiss Elyra S1 (Carl Zeiss) microscope equipped with a 63 × 1.4 NA oil objective lens and an Andor iXon 885 EM-CCD camera. Image acquisition was done with five grid rotations and image reconstruction was made with ZEN software. For QuASIMoDOH analysis, widefield images were used, with one ROI per cell selected to exclude edges. Background was removed by thresholding by Mode (*Paparelli et al., 2016*). QuASIMoDOH was run in batch for all ROIs. Statistical analysis of the inhomogeneity parameter was done in GraphPrism. For nearest-neighbor analysis and density measurement, image channels were aligned using Tetraspeck beads as fiducials in the ImageJ plugin B-unwarpJ. The removal of structured artifacts was done by manually adjusting the intensity histogram. Masks of intensity peaks were made by the ImageJ plugin H-watershed. Coordinates of particle centroids were extracted and used in a costume-written MATLAB routine to calculate the nearest-neighbor distance. To calculate random distances, we used the same set of images (e.g. NCT-PSEN1) but ROIs of one channel were paired with non-correspondent ROIs of the other channel.

## Photobleaching steps

As photobleaching steps require a low density of spots at the PM to discriminate one from another (*Ulbrich and Isacoff, 2007*), we used tKO NCT-SNAP/GFP-PSEN1 MEFs FAC Sorted for low levels of both tags. Cells were grown in 1.5 glass bottom Mattek dishes to 80% confluency and processed for PM sheets as described. PM sheets were fixed and imaged in PBS$^{-/-}$ on a home-build inverted wide-field setup equipped with an EM-CCD camera (full description in the PALM section). GFP was imaged in TIRF, recording for 2 min at 0.1 s/frame while photobleaching with 0.12 mW of 488 nm laser. Background subtraction was done in ImageJ by a sliding paraboloid with a rolling ball radius of 50px. One region of interest was analyzed per cell to avoid edge effects. We used the spot detection plugin of ImageJ, which averaged the first 10 frames, a 2px radius and noise thresholding of 200 to select spots and get intensity traces over time. Photobleaching steps were determined manually. An average of 64 spots was analyzed per cell (n = 4).

## Molecular Counting by Photon Statistics (CoPS)

PM sheets were imaged in a confocal setup Microtime 200 (PicoQuant) using an inverted microscope (IX73, Olympus) equipped with an oil objective lens (UPlanSApo 100 × 1.4 NA, Olympus). A fiber-coupled pulsed diode laser (LDH-D-C-485, PicoQuant) was used for 485 nm excitation at 10 MHz repetition rate. Fluorescence was separated from the excitation light by a dichroic mirror (Zt488/640rpc, Chroma) and split into four fractions by three 50:50 beam splitters (Beck Optronic Solutions Limited). Additional fluorescence filters (LP488, Semrock, and SP750, Chroma) were used in front of each of the four SPAD detectors (SPCM-AQRH-14-TR, Excelitas). Data were acquired with a time-correlated single-photon counting unit (HydraHarp 400, PicoQuant). Images were further analyzed with costume-written MATLAB routines (The Mathworks Inc).

## Live-cell single-molecule imaging and tracking for molecule counting

Single-molecule imaging was done live in GFP-PSEN1 or NCT-SNAP/GFP-PSEN1 in rescued PSEN1 sKO or PSEN1 and 2/NCT tKO MEFs, respectively, using Total Internal Reflection (TIRF) Microscopy on an Olympus IX71 microscope equipped with a 100 × 1.7 N.A. objective lens. Despite near-physiological levels of expression, levels were higher than the density limit of 2 spots/μm$^2$ for single-molecule imaging. Therefore, a circular region of 10–12 μm diameter in the center of the imaging field was photobleached using a focused laser steered by a set of scanning mirrors, similar to the Photo-Gate approach (*Madl et al., 2010*; *Belyy et al., 2017*). After the initial photobleaching, several rings were drawn with the focused laser at the edge of the bleached region at intervals of 5–10 s to control the re-population with unbleached molecules. For imaging, an iris in the TIRF illumination pathway was closed to eliminate glare from outside of the central region. Movies of 700 frames were recorded at 34 Hz with a back-illuminated EMCCD camera (Andor iXon DU-897).

### Evaluation of single-molecules intensity histograms

Tracking was done with the particle tracking tool from the MOSAIC suite in ImageJ, and intensities were extracted from the identified spots with a background subtraction. Spot intensities in the last 100 frames of the movie, where most GFP was photobleached, yielded the intensity distribution for complexes with a single GFP. The theoretical distribution of a complex containing two fluorescent GFPs was obtained by a convolution of the 1-GFP distribution with itself. The initial intensity distribution was extracted from frames 1–30 of the movie and fitted with a linear combination of the 1-GFP and 2-GFP distributions. To obtain the fractions of dimers and monomers, a fraction of 20% non-fluorescent GFP was taken into account (*Ulbrich and Isacoff, 2007*).

### Photo-Activated Localization Microscopy (PALM)

Intact cells stably rescued with mEOS3.2-PSEN1 and PM sheets derived thereafter were used for PALM. Imaging was done in TIRF mode in a home-build wide-field setup with an inverted microscope equipped with Nikon Perfect Focus System, a $100 \times 1.49$ NA APO-TIRF objective lens, an EM-CCD camera (9100-23B; Hamamatsu) and a 1.6x expansion lens (100 nm pixel size). During recording, the 405 nm activation laser intensity was manually adjusted to obtain the proper density of single-molecules per frame. Localizer software (*Dedecker and Neely, 2012*) was used for image reconstruction and extraction of coordinates. Cluster analysis by DBSCAN was done in qSR analysis platform (*Andrews, 2017*) with a 100 nm distance and a minimum of 10 localizations per cluster. An average of 80 clusters were analyzed per cell (n = 2 cells for entire cells, n = 4 cells for PM sheets).

### Single-Particle Tracking PALM (sptPALM)

Imaging was done using mEOS3.2-PSEN1 rescued cell lines. Cells were plated on cleaned coverslips 24 hr prior to imaging. On the day of imaging, cells were refreshed with complete medium supplemented with DMSO (VWR, 1:1000), DAPT (Tocris Bioscience, 1 µM) or L-685,458 (Calbiochem, 1 µM), and incubated for 1 hr. Coverslips were briefly washed with HBSS$^{+/+}$ and mounted on a holder in the presence with DMSO or inhibitors diluted in HBSS$^{+/+}$. The chamber was inserted on an inverted microscope Nikon Ti Eclipse (Nikon France S.A.S., Champigny-sur-Marne, France) equipped with a Perfect Focus System (PFS), a motorized stage TI-S-ER, and an azymuthal Ilas$^2$ TIRF arm (Gataca Systems, Massy, France) coupled to a laser bench containing 405 nm (100 mW), 491 nm (150 mW), 532 nm (1W), 561 nm (200 mW) and 642 nm (1W) diodes. Images were done using objective Apo TIRF 100 X oil NA 1.49 and a sensitive Evolve EMCCD camera (Photometrics, Tucson, USA). Photo-activation of mEOS3.2 was done using the Ilas$^2$ scanner system and 405 nm laser diode. The 37°C atmosphere was created with an incubator box and an air heating system (Life Imaging Services, Basel, Switzerland). This system was controlled by MetaMorph software (Molecular Devices, Sunnyvale, USA). Twenty cycles of 50 images at 0.05 s per frame were taken with a 2.5 s delay between each cycle. During SPT acquisitions, tracking based on wavelet segmentation were done using WaveTracer a MetaMorph (Molecular Devices, Sunnyvale, USA) add-on developed at the Interdisciplinary Institute of Neuroscience (IINS -UMR5297 -CNRS/University of Bordeaux) (*Kechkar et al., 2013*). Reconstruction of tracks and calculation of mean square displacement (MSD), diffusion coefficient and α coefficient were done using PALMTracer software, a MetaMorph (Molecular Devices, Sunnyvale, USA) add-on developed at the Interdisciplinary Institute of Neuroscience (IINS -UMR5297 -CNRS/University of Bordeaux), by Corey Butler (IINS -UMR5297 -CNRS/University of Bordeaux), Adel Mohamed Kechkar (Ecole Nationale Supérieure de Biotechnologie, Constantine, Algeria) and Jean-Baptiste Sibarita (IINS -UMR5297 -CNRS/University of Bordeaux). Briefly, single-molecule localization was achieved using wavelet segmentation, and then filtered out based on the quality of a 2D gaussian fit. SPT analysis was then done based on the detections using reconnection algorithms and for MSD and D calculations on reconnected trajectories (*Racine et al., 2007*; *Izeddin et al., 2012*; *Kechkar et al., 2013*). The instantaneous diffusion coefficient was calculated for each trajectory from a linear fit of the first four points of the MSD. The type of motility was inferred by the range of the α coefficient given the MSD fit $r^2 = t^{\alpha} + k$ (*Sibarita, 2014*; *Manzo and Garcia-Parajo, 2015*). Immobile tracks (D $\leq$ 0.01 µm²/s) were excluded, confined diffusion was defined as α <0.1, anomalous as 0.1$\leq$α $\leq$0.9, Brownian as 0.9$\leq$α $\leq$1.1, and directed as α >1.1 (*Sibarita, 2014*). The extraction of tracking data was done in MATLAB (The MathWorks, Inc).

## Single-particle tracking hotspot analysis

Hotspot analysis was done on the localizations of sptPALM data processed by Thunderstorm (*Ovesný et al., 2014*) with a wavelet of 3px. Four to eight ROIs of the same size were made per cell wherein cell edges were avoided in the quantification, and localization tables exported per ROI. Each ROI was analyzed with DBSCAN in the SR Tesseler (*Levet et al., 2015*) platform with a distance of 80 nm, a min localization density of 20 and minimum localizations per hotspot of 18 to account for more than two short tracks, which are the most abundant. The number of tracks per hotspot was measured manually.

## Statistical analysis

All statistical analysis was done in GraphPad Prism 7 or eight software (GraphPad, San Diego, CA). Statistical details for each particular experiment are described in the corresponding figure legends.

## Acknowledgements

The work in this manuscript was funded through VIB, KU Leuven (IDO/12/020, to WA and HM; C16/15/073 to WA), the FWO (to WA: S006617N, G078117N and G056017N), the FWO-Hercules foundation (to WA: AKUL/08/58, -/11/30, and -/13/39) and SAO-FRA (S#14017). The VIB Bioimaging Core acknowledges the Flemish government for acquiring the Zeiss ElyraS.1. SptPALM was done in the Bordeaux Imaging Center (BIC) a service unit of the CNRS-INSERM and Bordeaux University, member of the national infrastructure France BioImaging supported by the French National Research Agency (ANR-10-INBS-04). AE-A was a recipient of a short-term EMBO fellowship (#8358) to visit the BIC. The authors thank Pieter Baatsen of the VIB-EM core facility for help with scanning EM, Christel Poujol from BIC; Paja Reisch (PicoQuant) for measurements; Haisen Ta (PicoQuant) and Kiran Kumar Vudya Sethu for MATLAB scripts; Z Debyser for the plasmid pCHMWS-GFP-ires-hygro; the VIB-KU Leuven FACS facility core,Sam Lismont, Christine Michiels and David Demedts for technical support; Jean-Baptiste Sibarita (Bordeaux University), Jesse Aaron from the Advanced Imaging Center (AIC) and Chris Obara of the Lippincott-Schwartz lab (both at Janelia Research campus), Susana Rocha and Peter Dedecker (Science & Technology, KU Leuven), Marine Bretou and Rosanne Wouters for constructive discussions and members of the Annaert lab for critically reading of the manuscript.

## Additional information

### Competing interests

Caroline Berlage: Associated to PicoQuant. Marcelle Koenig: Works for PicoQuant. The other authors declare that no competing interests exist.

### Funding

| Funder | Grant reference number | Author |
|---|---|---|
| KU Leuven | IDO/12/020 | Hideaki Mizuno<br>Wim Annaert |
| KU Leuven | C16/15/073 | Wim Annaert |
| Fonds Wetenschappelijk Onderzoek | S006617N | Wim Annaert |
| Fonds Wetenschappelijk Onderzoek | G078117N | Wim Annaert |
| Fonds Wetenschappelijk Onderzoek | G056017N | Wim Annaert |
| Fonds Wetenschappelijk Onderzoek | AKUL/08/58 | Wim Annaert |
| Fonds Wetenschappelijk Onderzoek | AKUL/11/30 | Wim Annaert |

| Fonds Wetenschappelijk On-derzoek | AKUL/13/39 | Wim Annaert |
|---|---|---|
| Agence Nationale de la Re-cherche | ANR-10-INBS-04 | Magali Mondin |
| European Molecular Biology Organization | #8358 | Abril Angélica Escamilla-Ayala |
| SAO-FRA | #14017 | Wim Annaert |

The funders had no role in study design, data collection and interpretation, or the decision to submit the work for publication.

## Author contributions

Abril Angélica Escamilla-Ayala, Conceptualization, Data curation, Software, Formal analysis, Validation, Investigation, Visualization, Methodology, Writing - original draft, Writing - review and editing; Ragna Sannerud, Conceptualization, Formal analysis, Supervision, Investigation, Methodology, Writing - original draft; Magali Mondin, Resources, Formal analysis, Supervision, Methodology, Writing - original draft, Writing - review and editing; Karin Poersch, Caroline Berlage, Formal analysis, Investigation; Wendy Vermeire, Investigation, Methodology; Laura Paparelli, Software, Investigation, Methodology; Marcelle Koenig, Software, Methodology, Writing - original draft; Lucia Chavez-Gutierrez, Formal analysis, Methodology; Maximilian H Ulbrich, Formal analysis, Investigation, Methodology, Writing - original draft, Writing - review and editing; Sebastian Munck, Resources, Data curation, Formal analysis, Supervision, Methodology, Writing - original draft; Hideaki Mizuno, Conceptualization, Resources, Supervision, Writing - original draft, Writing - review and editing; Wim Annaert, Conceptualization, Supervision, Funding acquisition, Writing - original draft, Project administration, Writing - review and editing

## Author ORCIDs

Abril Angélica Escamilla-Ayala (iD) https://orcid.org/0000-0002-5761-0999
Maximilian H Ulbrich (iD) http://orcid.org/0000-0001-8123-1668
Wim Annaert (iD) https://orcid.org/0000-0003-0150-9661

## Decision letter and Author response

Decision letter https://doi.org/10.7554/eLife.56679.sa1
Author response https://doi.org/10.7554/eLife.56679.sa2

# Additional files

## Supplementary files

• Transparent reporting form

## Data availability

All data generated or analysed during this study are included in the manuscript and supporting files. Source data files have been provided for Figures 1, 2, 3 and 4.

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
