## [Decision Letter]

**Acceptance summary:**

Gamma-secretase complexes are critical in several diseases including Alzheimers. Most studies that characterised this multisubunit complex were based on fractionation and biochemistry. This study is unique in visualising single protein movements in the membrane and finds there are no preformed γ-secretase-substrate-mega-dalton complexes. This indicates the importance of membrane trafficking in the regulation of the activity of this protein complex.

**Decision letter after peer review:**

Thank you for submitting your article "Super-resolution microscopy reveals majorly monodisperse presenilin1/γ-secretase at the cell surface" for consideration by *eLife*. Your article has been reviewed by two peer reviewers, and the evaluation has been overseen by a Reviewing Editor and Olga Boudker as the Senior Editor. The following individual involved in review of your submission has agreed to reveal their identity: Sangram Sisodia (Reviewer #2).

The reviewers have discussed the reviews with one another and the Reviewing Editor has drafted this decision to help you prepare a revised submission.

Summary:

This paper by the Annaert group analyses how γ-secretase complexes move in the membrane. This is important as location of this complex defines function. Using advanced imaging technology, they show that secretase is largely monodisperse and very mobile but also that it coalesces in hotspots.

The reviewers concur that this paper is interesting and technically advanced, but descriptive and that it leaves a number of mechanistic questions to be answered (see comments reviewer 1). They ask this to be better acknowledged in the Discussion:

1) State that the mechanisms underlying several of the observations have not been resolved.

2) Where possible, speculate in the Discussion on possible underlying mechanisms that would explain the results. The reviewers also ask to tone down the conclusion that γ-secretase is monodisperse as the data suggest that only 30-35% is. This can be addressed with textual adaptations.

Finally, reviewer 1 also lists a number of issues that we believe can be addressed easily by textual changes or adaptations of figures:

Reviewer #1:

In addition to these overall comments, there are a number of specific items that should be attended to:

1) Was the confocal slice illustrated in Figure 1E taken in the middle of the cell? Specify. The authors state that many of the features detected are in endosomes, but no evidence is presented to support this statement. Were they co-localized with specific endosomal markers? If so, state which one(s) and show data.

2) In Figure 1I there are clearly two populations, one of which seems to overlap well with the theoretical random distribution. What is this second population and why is it not discussed explicitly, as it seems to encompass a large fraction of the molecules. Is it consistent with the statement thatPSEN1 and SNAP are associated? In Figure 1 and elsewhere please state the length of the size bars in the images.

3) Why are the clusters that are so clearly visible in Figure 1—figure supplement 2B not seen in Figure 2A?

4) The PhotoGate approach assumes that 100% of the molecules are mobile. Was this validated independently?

5) What explains the majority of single intensity features in the NCT-GFP/ GFP-PSEN1 experiments in Figure 2C-E?

6) What does the pseudocolor coding represent in Figure 3: is it individual particles or the time course of each molecule? Please clarify. Is the movement oriented in a particular direction, e.g. along microtubules or actin or vimentin fibers?

7) Could the failure of secretase to colocalize with ADAM10 and BACE1 be the result of overexpression of the latter, which unlike the other molecules found to colocalize, were expressed heterologously?

Revisions expected in follow-up work:

We believe it would be interesting to resolve the molecular basis behind the observations reported. The directed motion is interesting, but no information is provided regarding what drives it. Are molecular motors responsible? If so, are they microtubule- or actin-associated motors? And what is the consequence of such mobility on secretase function? Similarly, what underlies the formation of hotspots and what are their functional implications?

---

## [Author Response]

Revisions for this paper:The reviewers concur that this paper is interesting and technically advanced, but descriptive and that it leaves a number of mechanistic questions to be answered (see comments reviewer 1). They ask this to be better acknowledged in the Discussion:1) State that the mechanisms underlying several of the observations have not been resolved.2) Where possible, speculate in the Discussion on possible underlying mechanisms that would explain the results. The reviewers also ask to tone down the conclusion that γ secretase is monodisperse as the data suggest that only 30-35% is. This can be addressed with textual adaptations.Finally, reviewer 1 also lists a number of issues that we believe can be addressed easily by textual changes or adaptations of figures:Reviewer #1:In addition to these overall comments, there are a number of specific items that should be attended to:1) Was the confocal slice illustrated in Figure 1E taken in the middle of the cell? Specify. The authors state that many of the features detected are in endosomes, but no evidence is presented to support this statement. Were they co-localized with specific endosomal markers? If so, state which one(s) and show data.

The images were taken at the lower half of the cell to be able to visualize the ruffles at the edge of the plasma membrane. We have previously published (Sannerud et al., 2016) that PSEN1 complexes are residing at the cell surface as well as in endosomal compartments, including early, recycling, and smaller amounts in late endosomes. To illustrate this, we have extended the same sample as displayed in Figure 1E with the co-localization of PSEN1/NCT with LAMP1 (new panel H in Figure 1—figure supplement 1; arrowheads in S1-H). We routinely use this marker as it is well established that LAMP1 traffics via the cell surface and endosomal compartments to LE/Lys vesicles. Also, in Figure 1E, we have additionally highlighted the co-localization of PSEN1-NCT at the cell surface (yellow arrowheads) and membrane ruffles (orange arrowheads). We have clarified this also in more detail in the legends to the figure and in Results section:

“Confocal imaging revealed co-localized subunits at the cell surface, including membrane ruffles (Figure 1E), and in LAMP1-positive organelles (Figure 1—figure supplement 1H).”2) In Figure 1I there are clearly two populations, one of which seems to overlap well with the theoretical random distribution. What is this second population and why is it not discussed explicitly, as it seems to encompass a large fraction of the molecules. Is it consistent with the statement thatPSEN1 and SNAP are associated?

We agree with this observation, on a second population being closer to a random distribution. In the case of the GFP-PSEN1/GFPnb pair this is a rather small population (16% of all), while for the GFP-PSEN1/NCT-SNAP pair this is slightly higher (30% of all). For both, these represents (minor) fractions with much larger distances of either GFPnb or NCT-SNAP to GFP-PSEN1. They likely correspond to GFPnb or NCT-SNAP that pair with GFP-PSEN1 of another complex that is not visible either due to GFP photobleaching or immature chromophores as reported (Ulbrich and Isacoff, 2007): herein, the expected fraction of fluorescent GFP at the PM for any given GFP construct is ~80%, which is in line with the ~80% of association either of GFPnb or NCT-SNAP to GFP-PSEN1. We have clarified this as such now in the text of the revised manuscript, Results section:

“We noticed a fraction of the experimental population that follows the random distribution in both cases, representing about 16% for GFPnb and 30% for NCT of the total population. These likely correspond to GFPnb or NCT-SNAP that pair with GFP-PSEN1 of another complex that is not visible, either due to GFP photobleaching or immature chromophores. Indeed, the expected fraction of fluorescent GFP at the PM for a given GFP construct is ~80%, which is in line with our observed association to GFPnb or NCT-SNAP by GFP-PSEN1 (Ulbrich and Isacoff, 2007).”

In Figure 1 and elsewhere please state the length of the size bars in the images.

We have added the length of the size bars in the respective panels of Figure 1 as well as throughout all other figures.

3) Why are the clusters that are so clearly visible in Figure 1—figure supplement 2B not seen in Figure 2A?

We thank the reviewer for this comment and there is a very logic explanation for this as the microscopy techniques used in both panels are very different. In panel 2A, we are using structured Illumination microscopy (SIM) on GFP. Here, all GFP molecules are excited at once during five grid rotations, after which, mathematical reconstruction allows to get 80-100nm lateral resolution. However, in panel S2B single-molecule localization microscopy (SMLM) is used, which is based in the repetitive detection of stochastically activated single molecules through many frames. After centroid localization of each detection, all detections are pulled to a single image. This creates artificial clusters, as widely reported, and especially for mEOS3.2 (Annibale et al., 2010, 2011). Many strategies have been explored to minimize the artifacts created by the multi-detection of a single molecule. However, these calculations are complex and labor extensive, thus we only used SMLM as control for PM sheet distribution versus entire cells. For the actual molecular counting of PSEN1, we preferred to use the for GFP well-established photobleaching and photon-counting techniques. Additional information was added to the Results section:

*“*Of note, these clusters shown by PALM do not represent an equal number of mEOS3.2-PSEN1 molecules but result from the blinking property of mEOS3.2 giving rise to multiple localizations per molecule.”

4) The PhotoGate approach assumes that 100% of the molecules are mobile. Was this validated independently?

In the PhotoGate approach, the central area is initially completely photobleached and then repopulated by diffusion from outside the central area. Therefore, only mobile spots can be analyzed. Together with the later result from Figure 3 where we show that about 20% of spots are immobile, the conclusions from the PhotoGate approach apply at least for the major fraction of 80% of total molecules. It is not excluded that the immobile fraction follows a different intensity distribution. We clarify this point further in the Results section:

“Hence, this method only allows us to study mobile complexes. Since these mobile complexes were not exposed…”

5) What explains the majority of single intensity features in the NCT-GFP/ GFP-PSEN1 experiments in Figure 2C-E?

We deduced the fraction of spots with double intensity by assuming the final intensity distribution (after photobleaching) is solely based on 1-GFP spots, and convolving this distribution with itself, which gives an estimate of the intensity distribution of spots containing 2 GFPs. A fit to the observed distribution before photobleaching yielded a fraction of 58% of 1-GFP spots and 42% of 2-GFP spots. In previous work, a 80% fraction of functional GFP and 20% non-functional GFP were observed, which results in 33% of 1-GFP spots and 67% of 2-GFP spots in a pure dimer (Ulbrich and Isacoff, 2007). The higher prevalence of 1-GFP spots in our experiments can be explained either by pre-bleaching during searching a cell and adjustment of the focus (leaving 59% of functional GFP), by 41% monomeric subunits (containing only 1 GFP, i.e. either NCT-GFP or GFP-PSEN1) , or a combination of both. Overall literature agrees that, in post-Golgi compartments, g-secretase subunits exist essentially in a complex. Therefore, the fraction of monomeric subunits that cannot be explained as a result of prebleaching events might originate from other pools such as the cortical ER, that is known to very closely appose to the PM. The ER harbors relatively most of the unassembled subunits further supporting this. We clarify this point in the text and adjusted Figure 2E to show the two populations of 1-GFP spots and 2-GFP spots more clearly. Adjustments to the text can be found in the Results section:

“We compared intensity histograms of the beginning and the end of the video, after which most GFPs are bleached and therefore, only complexes with a single functional GFP should be left (Figure 2E). […] The ER harbors relatively most of the unassembled and immature subunits, further supporting this.”

6) What does the pseudocolor coding represent in Figure 3: is it individual particles or the time course of each molecule? Please clarify.

The pseudocolor indeed represent individual tracks at different time points or frames where the track started. To clarify this better we have included color bars were needed.

Is the movement oriented in a particular direction, e.g. along microtubules or actin or vimentin fibers?

This is an interesting point. So far, we have not found evidence that the overall movement of PSEN/γ-secretase is oriented in a particular direction. We elaborated further in the Discussion and speculated on possible mechanisms that could drive PSEN1/γ-secretase diffusion, specifically regarding directed motion. As molecular motors are unlikely to act on plasma membrane protein diffusion, alternative explanations are provided through the “uniform flow” and the “conveyor belt” models. In the first one, particles diffuse with the membrane in a constant flow. Whereas in the conveyor belt, particles switch to direct diffusion transiently when “hopping on” the conveyor belt. This could be the underlying actin cytoskeleton for instance. In this regard, PSEN1 has indeed been reported to interact with filamin, which could anchor PSEN1 to actin filaments. Likewise, using the moss Physcomitrella patens as a model organism, the group of Rafi Kopan has linked PSEN function to the cytoskeleton. They found that in absence of PS, there were growth abnormalities related to cytoskeleton defects, pointing to a scaffolding role of PS of the cytoskeleton with adhesion molecules (Khandelwal et al., 2007). We have preliminary data that show an effect of latrunculinA on PSEN1 motility but these are currently too preliminary to include and with further experimentation, we will address this in more detail in a follow-up paper that could be considered as a Research Advance in *eLife*. Discussion about the underlying mechanisms for directed movement can be found in Discussion section:

“SptPALM revealed that PSEN1/γ-secretase changes motility during its diffusion, with frequent intervals of directed movement (Figure 3C). […] To explore directed motility of PSEN1/γ-secretase in relation to the actin cytoskeleton, more appropriate cellular models including migrating cells, or the neuronal growth cone might be considered.”

7) Could the failure of secretase to colocalize with ADAM10 and BACE1 be the result of overexpression of the latter, which unlike the other molecules found to colocalize, were expressed heterologously?

This an important point raised by the reviewer. For γ-secretase subunits, we systematically use stable lentiviral rescue in the respective KO backgrounds followed by selection for low(est) expressing clones (as verified with antibodies and compared to endogenous wild-type expressions). As for APP and Cadherin, we first tried to use primary antibodies directed against the sheddases, but after testing many of the reported antibodies, for instance in ADAM10 KO cell lines, we realized that they all give a lot of unspecific labeling. Without the availability of antibodies with a high signal to noise ratio, it makes it merely impossible to perform reliable nearest neighbor distance analysis. As opposed to γ-secretase that requires stoichiometric expression, this is not the case for sheddases making exogenous expression using SNAP-tagged versions an acceptable alternative. Herein, we systematically imaged only the lowest expressing cells, exactly to avoid overexpression artifacts.

Of note, if ADAM10 or BACE1 should be higher overexpressed compared to endogenous, we argue that in this case, rather the opposite might occur: if the macrocomplex idea is true, we would expect to rather see an overlap with PSEN1/γ-secretase as they would outcompete the endogenous sheddases. As this is not observed, we are confident that our data support our current interpretation that PSEN1/γ-secretase is not being associated with the respective sheddases. This was further corroborated with our sptPALM analysis (Figure 5D,F), rather supporting the idea that there might be temporarily regulated encounters in particular areas where sheddases reside and may deliver the CTF for further processing by γ-secretase. To clarify this point we added some text in Results section:

“Of note, because of the lack of proper antibodies, we expressed SNAP-tagged ADAM10 or BACE1 in GFP-PSEN1 expressing sKO MEFs and limited our analysis only to those cells with the lowest expression to avoid overexpression artifacts.”

**References**

Annibale, P., M. Scarselli, A. Kodiyan, and A. Radenovic. 2010. Photoactivatable Fluorescent Protein mEos2 Displays Repeated Photoactivation after a Long-Lived Dark State in the Red Photoconverted Form. *J. Phys. Chem. Lett.* 1:1506–1510. doi:10.1021/jz1003523.Annibale, P., S. Vanni, M. Scarselli, U. Rothlisberger, and A. Radenovic. 2011. Identification of clustering artifacts in photoactivated localization microscopy. *Nat. Methods*. 8:527–528. doi:10.1038/nmeth.1627.